# Tempo Adaptation in Non-stationary Reinforcement Learning

**Hyunin Lee**[1,*]    **Yuhao Ding**[1]    **Jongmin Lee**[1]    **Ming Jin**[2,*]
**Javad Lavaei**[1]    **Somayeh Sojoudi**[1]
[1]UC Berkeley, Berkeley, CA 94709
[2]Virginia Tech, Blacksburg, VA 24061
{hyunin,yuhao_ding,jongmin.lee,lavaei,sojoudi}@berkeley.edu
jinming@vt.edu

## Abstract

We first raise and tackle a "time synchronization" issue between the agent and the environment in non-stationary reinforcement learning (RL), a crucial factor hindering its real-world applications. In reality, environmental changes occur over wall-clock time ($t$) rather than episode progress ($k$), where wall-clock time signifies the actual elapsed time within the fixed duration $t \in [0, T]$. In existing works, at episode $k$, the agent rolls a trajectory and trains a policy before transitioning to episode $k + 1$. In the context of the time-desynchronized environment, however, the agent at time $t_k$ allocates $\Delta t$ for trajectory generation and training, subsequently moves to the next episode at $t_{k+1} = t_k + \Delta t$. Despite a fixed total number of episodes ($K$), the agent accumulates different trajectories influenced by the choice of *interaction times* ($t_1, t_2, ..., t_K$), significantly impacting the suboptimality gap of the policy. We propose a Proactively Synchronizing Tempo (`ProST`) framework that computes a suboptimal sequence $\{t_1, t_2, ..., t_K\}(= \{t\}_{1:K})$ by minimizing an upper bound on its performance measure, i.e., the dynamic regret. Our main contribution is that we show that a suboptimal $\{t\}_{1:K}$ trades-off between the policy training time (agent tempo) and how fast the environment changes (environment tempo). Theoretically, this work develops a suboptimal $\{t\}_{1:K}$ as a function of the degree of the environment's non-stationarity while also achieving a sublinear dynamic regret. Our experimental evaluation on various high-dimensional non-stationary environments shows that the `ProST` framework achieves a higher online return at suboptimal $\{t\}_{1:K}$ than the existing methods.

## 1 Introduction

The prevailing reinforcement learning (RL) paradigm gathers past data, trains models in the present, and deploys them in the *future*. This approach has proven successful for *stationary* Markov decision processes (MDPs), where the reward and transition functions remain constant [1–3]. However, challenges arise when the environments undergo significant changes, particularly when the reward and transition functions are dependent on time or latent factors [4–6], in *non-stationary* MDPs. Managing non-stationarity in environments is crucial for real-world RL applications. Thus, adapting to changing environments is pivotal in non-stationary RL.

This paper addresses a practical concern that has inadvertently been overlooked within traditional non-stationary RL environments, namely, the time synchronization between the agent and the environment. We raise the impracticality of utilizing *episode-varying* environments in existing non-stationary RL

---

* Corresponding authors. This work was supported by grants from ARO, ONR, AFOSR, NSF, and the UC Noyce Initiative.

37th Conference on Neural Information Processing Systems (NeurIPS 2023).

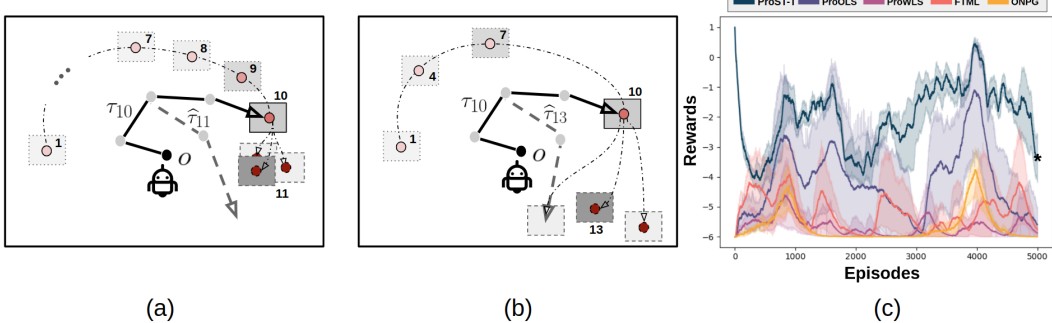

Figure 1: (a) 2D goal reacher in a time-desynchronized environment for one policy update, where the agent learns an inaccurate policy on an accurate model; (b) For three policy updates, the agent learns a near-optimal policy on an inaccurate model; (c) Rewards per episode in 2D goal reacher with four model-free baselines, where `ProST-T`$^*$ is one of our proposed methods.

research, as such environments do not align with the real-world scenario where changes occur regardless of the agent's behavior. In an episode-varying environment, the agent has complete control over determining the time to execute the episode $k$, the duration of policy training between the episodes $k$ and $k + 1$, and the transition time to the episode $k + 1$. The issue stems from the premise that the environment undergoes dynamic changes throughout the course of each episode, with the rate of non-stationarity contingent upon the behavior exhibited by the agent. However, an independent *wall-clock time* ($\mathfrak{t}$) exists in a real-world environment, thereby the above three events are now recognized as wall-clock time $\mathfrak{t}_k$, training time $\Delta\mathfrak{t}$, and $\mathfrak{t}_{k+1}$. The selection of interaction times $(\mathfrak{t}_k, \mathfrak{t}_{k+1})$ has a notable impact on the collected trajectories, while the interval $\mathfrak{t}_{k+1} - \mathfrak{t}_k$ establishes an upper limit on the duration of training ($\Delta\mathfrak{t}$). This interval profoundly influences the suboptimality gap of the policy. In the context of a time-desynchronized environment, achieving an optimal policy requires addressing a previously unexplored question: the determination of the *optimal time sequence* $\{\mathfrak{t}_1, \mathfrak{t}_2, ..., .\mathfrak{t}_K\}(= \{\mathfrak{t}\}_{1:K})$ at which the agent should interact with the environment.

We elucidate the significance of the aforementioned research question through an example. Consider a robot with the goal of reaching inside a gray-shaded non-fixed target box, known as the goal reacher (Appendix A.1). Note that the reward changes as the position of the box changes over time (Figure 1-(a)). We begin by considering a scenario in which the wall-clock time and episode are synchronized, wherein the environment evolves alongside the episode. During each episode $k$, the agent rollouts a trajectory and iteratively updates the policy $N$ times, with the assumption that one policy update requires one second, and then the agent transitions to the subsequent episode $k + 1$. In conventional non-stationary RL environments, it is evident that a larger value of $N$ provides an advantage in terms of a faster adaptation to achieve a near-optimal policy. However, regardless of the chosen value of $N$, the agent will consistently encounter the same environment in the subsequent episode. Now, consider a scenario in which the wall-clock time and episode are desynchronized. In this context, given a fixed wall-clock time duration $\mathfrak{t} \in [0, 10]$, the agent is faced with the additional task of determining both the total number of interactions (denoted as the total episode $K$) and the specific time instances for these interactions $\{\mathfrak{t}\}_{1:K}$, where $\mathfrak{t}_k \in [0, 10], \mathfrak{t}_{k-1} < \mathfrak{t}_k$ for $\forall k \in [K]$. Figure 1(a) shows an agent that interacts with the environment ten times, that is, $\{\mathfrak{t}\}_{1:K} = \{1, 2, ..., 10\}$, and spends the time interval $(\mathfrak{t}_k, \mathfrak{t}_{k+1})$ to train the policy, which consumes one second ($K = 10, N = 1$). The high frequency of interaction ($K = 10$) provides adequate data for precise future box position learning ($\mathfrak{t} = 11$), yet a single policy update ($N = 1$) may not approximate the optimal policy. Now, if the agent interacts with the environment four times, i.e. $\{\mathfrak{t}\}_{1:K} = \{1, 4, 7, 10\}$ (see Figure 1(b)), it becomes feasible to train the policy over a duration of three seconds ($K = 4, N = 3$). A longer period of policy training ($N = 3$) helps the agent in obtaining a near-optimal policy. However, limited observation data ($K = 4$) and large time intervals ($\mathfrak{t} \in \{11, 12, 13\}$) may lead to less accurate box predictions. This example underscores the practical importance of aligning the interaction time of the agent with the environment in non-stationary RL. Determining the optimal sequence $\{\mathfrak{t}\}_{1:K}$ involves a trade-off between achieving an optimal model and an optimal policy.

Based on the previous example, our key insight is that, in non-stationary environments, the new factor **tempo** emerges. Informally, tempo refers to the pace of processes occurring in a non-stationary

environment. We define **environment tempo** as how fast the environment changes and **agent tempo** as how frequently it updates the policy. Despite the importance of considering the tempo to find the optimal $\{t\}_{1:K}$, the existing formulations and methods for non-stationarity RL are insufficient. None of the existing works has adequately addressed this crucial aspect.

Our framework, `ProST`, provides a solution to finding the optimal $\{t\}_{1:K}$ by computing a minimum solution to an upper bound on its performance measure. It proactively optimizes the time sequence by leveraging the agent tempo and the environment tempo. The `ProST` framework is divided into two components: future policy optimizer ($\text{OPT}_\pi$) and time optimizer ($\text{OPT}_t$), and is characterized by three key features: 1) it is *proactive* in nature as it forecasts the future MDP model; 2) it is *model-based* as it optimizes the policy in the created MDP; and 3) it is a *synchronizing tempo* framework as it finds a suboptimal training time by adjusting how many times the agent needs to update the policy (agent tempo) relative to how fast the environment changes (environment tempo). Our framework is general in the sense that it can be equipped with any common algorithm for policy update. Compared to the existing works [7–9], our approach achieves higher rewards and a more stable performance over time (see Figure 1(c) and Section 5).

We analyze the statistical and computational properties of `ProST` in a tabular MDP, which is named `ProST-T`. Our framework learns in a novel MDP, namely elapsed time-varying MDP, and quantifies its non-stationarity with a novel metric, namely time-elapsing variation budget, where both consider wall-clock time taken. We analyze the dynamic regret of `ProST-T` (Theorem 1) into two components: dynamic regret of $\text{OPT}_\pi$ that learns a future MDP model (Proposition 1) and dynamic regret of $\text{OPT}_t$ that computes a near-optimal policy in that model (Theorem 2, Proposition 2). We show that both regrets satisfy a sublinear rate with respect to the total number of episodes regardless of the agent tempo. More importantly, we obtain suboptimal training time by minimizing an objective that strikes a balance between the upper bounds of those two dynamic regrets, which reflect the tempos of the agent and the environment (Theorem 3). We find an interesting property that the future MDP model error of $\text{OPT}_\pi$ serves as a common factor on both regrets and show that the upper bound on the dynamic regret of `ProST-T` can be improved by a joint optimization problem of learning both different weights on observed data and a model (Theorem 4, Remark 1).

Finally, we introduce `ProST-G`, which is an adaptable learning algorithm for high-dimensional tasks achieved through a practical approximation of `ProST`. Empirically, `ProST-G` provides solid evidence on different reward returns depending on policy training time and the significance of learning the future MDP model. `ProST-G` also consistently finds a near-optimal policy, outperforming four popular RL baselines that are used in non-stationary environments on three different Mujoco tasks.

**Notation**

The sets of natural, real, and non-negative real numbers are denoted by $\mathbb{N}, \mathbb{R}$, and $\mathbb{R}_+$, respectively. For a finite set $Z$, the notation $|Z|$ denotes its cardinality and the notation $\Delta(Z)$ denotes the probability simplex over $Z$. For $X \in \mathbb{N}$, we define $[X]:=\{1, 2, .., X\}$. For a variable $X$, we denote $\widehat{X}$ as a *forecasted* (or *predicted*) variable at the current time, and $\widetilde{X}$ as the observed value in the past. Also, for any time variable $t > 0$ and $k \in \mathbb{N}$, we denote the time sequence $\{t_1, t_2, .., t_k\}$ as $\{t\}_{1:k}$, and variable $X$ at time $t_k$ as $X_{t_k}$. We use the shorthand notation $X_{(k)}$(or $X^{(k)}$) for $X_{t_k}$(or $X^{t_k}$). We use the notation $\{x\}_{a:b}$ to denote a sequence of variables $\{x_a, x_{a+1}, ..., x_b\}$, and $\{x\}_{(a:b)}$ to represent a sequence of variables $\{x_{t_a}, x_{t_{a+1}}, ..., x_{t_b}\}$. Given two variables $x$ and $y$, let $x \vee y$ denote $\max(x,y)$, and $x \wedge y$ denote $\min(x,y)$. Given two complex numbers $z_1$ and $z_2$, we write $z_2 = W(z_1)$ if $z_2 e^{z_2} = z_1$, where $W$ is the Lambert function. Given a variable $x$, the notation $a = \mathcal{O}(b(x))$ means that $a \leq C \cdot b(x)$ for some constant $C > 0$ that is independent of $x$, and the notation $a = \Omega(b(x))$ means that $a \geq C \cdot b(x)$ for some constant $C > 0$ that is independent of $x$. We have described the specific details in Appendix C.1.

## 2   Problem statement: Desynchronizing timelines

### 2.1   Time-elapsing Markov Decision Process

In this paper, we study a non-stationary Markov Decision Process (MDP) for which the transition probability and the reward change over time. We begin by clarifying that the term *episode* is agent-centric, not environment-centric. Prior solutions for episode-varying (or step-varying) MDPs operate

under the assumption that the timing of MDP changes aligns with the agent commencing a new episode (or step). We introduce the new concept of **time-elapsing MDP**. It starts from the wall-clock time $\mathsf{t} = 0$ to $\mathsf{t} = T$, where $T$ is fixed. The time-elapsing MDP at time $\mathsf{t} \in [0, T]$ is defined as $\mathcal{M}_\mathsf{t} \coloneqq \langle \mathcal{S}, \mathcal{A}, H, P_\mathsf{t}, R_\mathsf{t}, \gamma \rangle$, where $\mathcal{S}$ is the state space, $\mathcal{A}$ is the action space, $H$ is the number of steps, $P_\mathsf{t} : \mathcal{S} \times \mathcal{A} \times \mathcal{S} \to \Delta(\mathcal{S})$ is the transition probability , $R_\mathsf{t} : \mathcal{S} \times \mathcal{A} \to \mathbb{R}$ is the reward function, and $\gamma$ is a discounting factor. Prior to executing the first episode, the agent determines the total number of interactions with the environment (denoted as the number of total episode $K$) and subsequently computes the sequence of interaction times $\{\mathsf{t}\}_{1:K}$ through an optimization problem. We denote $\mathsf{t}_k$ as the wall-clock time of the environment when the agent starts the episode $k$. Similar to the existing non-stationary RL framework, the agent's objective is to learn a policy $\pi^{\mathsf{t}_k} : \mathcal{S} \to \Delta(\mathcal{A})$ for all $k$. This is achieved through engaging in a total of $K$ episode interactions, namely $\{\mathcal{M}_{\mathsf{t}_1}, \mathcal{M}_{\mathsf{t}_1}, ..., \mathcal{M}_{\mathsf{t}_K}\}$, where the agent dedicates the time interval $(\mathsf{t}_k, \mathsf{t}_{k+1})$ for policy training and then obtains a sequence of suboptimal policies $\{\pi^{\mathsf{t}_1}, \pi^{\mathsf{t}_2}, ..., \pi^{\mathsf{t}_K}\}$ to maximize a non-stationary policy evaluation metric, *dynamic regret*.

Dealing with time-elapsing MDP instead of conventional MDP raises an important question that should be addressed: within the time duration $[0, T]$, what time sequence $\{\mathsf{t}\}_{1:K}$ yields favorable trajectory samples to obtain an optimal policy? This question is also related to the following: what is optimal value of $K$, i.e. the total number of episode that encompasses a satisfactory balance between the amount of observed trajectories and the accuracy of policy training? These interwined questions are concerned with an important aspect of RL, which is the computation of the optimal policy for a given $\mathsf{t}_k$. In Section 4, we propose the `ProST` framework that computes a suboptimal $K^*$ and its corresponding suboptimal time sequence $\{\mathsf{t}^*\}_{1:K^*}$ based on the information of the environment's rate of change. In Section 3, we compute a near-optimal policy for $\{\mathsf{t}^*\}_{1:K^*}$. Before proceeding with the above results, we introduce a new metric quantifying the environment's pace of change, referred to as time-elapsing variation budget.

## 2.2 Time-elapsing variation budget

*Variation budget* [10–12] is a metric to quantify the speed with which the environment changes. Driven by our motivations, we introduce a new metric imbued with real-time considerations, named *time-elapsing variation budget* $B(\Delta \mathsf{t})$. Unlike the existing variation budget, which quantifies the environment's non-stationarity across episodes $\{1, 2, .., K\}$, our definition accesses it across $\{\mathsf{t}_1, \mathsf{t}_2, ..., \mathsf{t}_K\}$, where the interval $\Delta \mathsf{t} = \mathsf{t}_{k+1} - \mathsf{t}_k$ remains constant regardless of $k \in [K-1]$. For further analysis, we define two time-elapsing variation budgets, one for transition probability and another for reward function.

**Definition 1** (Time-elapsing variation budgets)**.** *For a given sequence* $\{\mathsf{t}_1, \mathsf{t}_2, .., \mathsf{t}_K\}$, *assume that the interval* $\Delta \mathsf{t}$ *is equal to the policy training time* $\Delta_\pi$. *We define two time-elapsing variation budgets* $B_p(\Delta_\pi)$ *and* $B_r(\Delta_\pi)$ *as*

$$B_p(\Delta_\pi) \coloneqq \sum_{k=1}^{K-1} \sup_{s,a} \|P_{\mathsf{t}_{k+1}}(\cdot \,|s,a) - P_{\mathsf{t}_k}(\cdot \,|s,a)\|_1, \; B_r(\Delta_\pi) \coloneqq \sum_{k=1}^{K-1} \sup_{s,a} |R_{\mathsf{t}_{k+1}}(s,a) - R_{\mathsf{t}_k}(s,a)|.$$

To enhance the representation of a real-world system using the time-elapsing variation budgets, we make the following assumption.

**Assumption 1** (Drifting constants)**.** *There exist constants* $c > 1$ *and* $\alpha_r, \alpha_p \geq 0$ *such that* $B_p(c\Delta_\pi) \leq c^{\alpha_p} B_p(\Delta_\pi)$ *and* $B_r(c\Delta_\pi) \leq c^{\alpha_r} B_r(\Delta_\pi)$. *We call* $\alpha_p$ *and* $\alpha_r$ *the drifting constants.*

## 2.3 Suboptimal training time

Aside from the formal MDP framework, the agent can be informed of varying time-elapsing variation budgets based on the training time $\Delta_\pi \in (0, T)$ even within the same time-elapsing MDP. Intuitively, a short time $\Delta_\pi$ is inadequate to obtain a near-optimal policy, yet it facilitates frequent interactions with the environment, leading to a reduction in empirical model error due to a larger volume of data. On the contrary, a long time $\Delta_\pi$ may ensure obtaining a near-optimal policy but also introduces greater uncertainty in learning the environment. This motivates us to find a **suboptimal training time** $\Delta_\pi^* \in (0, T)$ that strikes a balance between the sub-optimal gap of the policy and the empirical model error. If it exists, then $\Delta_\pi^*$ provides a suboptimal $K^* = \lfloor T/\Delta_\pi^* \rfloor$, and a suboptimal time sequence where $\mathsf{t}_k^* = \mathsf{t}_1 + \Delta_\pi^* \cdot (k-1)$ for all $k \in [K^*]$. Our `ProST` framework computes the parameter $\Delta_\pi^*$,

then sets $\{t^*\}_{1:K^*}$, and finally finds a *future* near-optimal policy for time $t^*_{k+1}$ at time $t^*_k$. In the next section, we first study how to approximate the one-episode-ahead suboptimal policy $\pi^{*,t_{k+1}}$ at time $t_k$ when $\{t\}_{1:K}$ is given.

## 3 Future policy optimizer

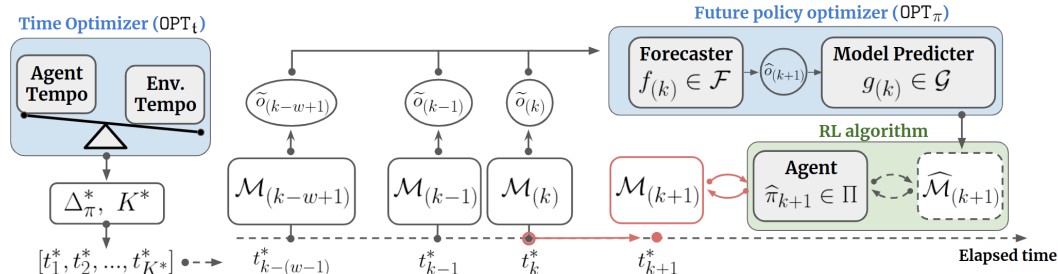

Figure 2: ProST framework

For given $t_k$ and $t_{k+1}$, the future policy optimizer ($\text{OPT}_\pi$), as a module of the ProST framework (Figure 2), computes a near-optimal policy for the future time $t_{k+1}$ at time $t_k$ via two procedures: (i) it first forecasts the future MDP model of time $t_{k+1}$ at time $t_k$ utilizing the MDP forecaster function, (ii) it then utilizes an arbitrary policy optimization algorithm within the forecasted MDP model $\text{OPT}_\pi$ to obtain a future near-optimal policy $\pi^{*,t_{k+1}}$.

### 3.1 MDP forecaster

Our ProST framework is applicable in an environment that meets the following assumption.

**Assumption 2** (Observable non-stationary set $\mathcal{O}$). *Assume that the non-stationarity of $\mathcal{M}_{t_k}$ is fully characterized by a non-stationary parameter $o_{t_k} \in \mathcal{O}$. Assume also that the agent observes a noisy non-stationary parameter $\tilde{o}_{t_k}$ at the end of episode $k \in [K]$ (at time $t_k$).*

It is worth noting that Assumption 2 is mild, as prior research in non-stationary RL has proposed techniques to estimate $o_{(k)}$ through latent factor identification methods [4, 13–16], and our framework accommodates the incorporation of those works for the estimation of $o_{(k)}$. Based on Assumption 2, we define the MDP forecaster function $g \circ f$ below.

**Definition 2** (MDP forecaster $g \circ f$). *Consider two function classes $\mathcal{F}$ and $\mathcal{G}$ such that $\mathcal{F} : \mathcal{O}^w \to \mathcal{O}$ and $\mathcal{G} : \mathcal{S} \times \mathcal{A} \times \mathcal{O} \to \mathbb{R} \times \Delta(\mathcal{S})$, where $w \in \mathbb{N}$. Then, for $f_{(k)} \in \mathcal{F}$ and $g_{(k)} \in G$, we define MDP forecaster at time $t_k$ as $(g \circ f)_{(k)} : \mathcal{O}^w \times \mathcal{S} \times \mathcal{A} \to \mathbb{R} \times \Delta(\mathcal{S})$.*

The function $f_{(k)}$, acting as a non-stationarity forecaster, predicts a non-stationary parameter $\hat{o}_{(k+1)}$ at time $t_{k+1}$ based on the last $w$ observations given by the set $\{\tilde{o}\}_{(k-w+1:k)}$, i.e., $\hat{o}_{(k+1)} = f(\{\tilde{o}\}_{(k-w+1,k)})$. The agent can determine the number of used historical observations, denoted as $w$, by leveraging information from the environment (Section 4). Then, the function $g_{(k)}$, acting as a model predictor, predicts a reward $\widehat{R}_{(k+1)}(s,a)$ and a transition probability $\widehat{P}_{(k+1)}(\cdot|s,a)$ for time $t_{k+1}$, i.e., $(\widehat{R}_{(k+1)}, \widehat{P}_{(k+1)}) = g_{(k)}(s,a,\hat{o}_{k+1})$. Finally, the $\text{OPT}_\pi$ generates the estimated future MDP $\widehat{\mathcal{M}}_{(k+1)} = \langle \mathcal{S}, \mathcal{A}, H, \widehat{P}_{(k+1)}, \widehat{R}_{(k+1)}, \gamma \rangle$ associated with time $t_{k+1}$.

### 3.2 Finding future optimal policy

Now, consider an arbitrary RL algorithm provided by the user to obtain an optimal policy from the model $\widehat{\mathcal{M}}_{(k+1)}$. For a given time sequence $\{t\}_{1:K}$, the $\text{OPT}_\pi$ finds a near-optimal future policy as follows: (1) observe and forecast, (2) optimize using the future MDP model.

**(1) Observe and forecast.** At time $t_k$, the agent executes an episode $k$ in the environment $\mathcal{M}_{(k)}$, completes its trajectory $\tau_{(k)}$, and observes the noisy non-stationary parameter $\hat{o}_{(k)}$ (Assumption 2). The algorithm then updates the function $f_{(k)}$ based on the last $w$ observed parameters, and the

function $g_{(k)}$ with input from all previous trajectories. Following these updates, the MDP forecaster at time $t_k$ predicts $\widehat{P}_{(k+1)}$ and $\widehat{R}_{(k+1)}$, thus creating the MDP model $\widehat{\mathcal{M}}_{(k+1)}$ for time $t_{k+1}$.

**(2) Optimize using the future MDP model.** Up until time $t_{k+1}$, the agent continually updates the policy within the estimated future MDP $\widehat{\mathcal{M}}_{(k+1)}$ for a given duration $\Delta_\pi$. Specifically, the agent rollouts synthetic trajectories $\hat{\tau}_{(k+1)}$ in $\widehat{\mathcal{M}}_{(k+1)}$, and utilizes any policy update algorithm to obtain a policy $\widehat{\pi}_{(k+1)}$. Following the duration $\Delta_\pi$, the agent stops the training by the time $t_{k+1}$ and moves to the next episode $\mathcal{M}_{(k+1)}$ with policy $\widehat{\pi}_{(k+1)}$.

We elaborate on the above procedure in Algorithm 1 given in Appendix F.1.

## 4 Time optimizer

### 4.1 Theoretical analysis

We now present our main theoretical contribution, which is regarding the time optimizer ($\mathtt{OPT_t}$): computing a suboptimal policy training time $\Delta_\pi^*$ (the agent tempo). Our theoretical analysis starts with specifying the components of the $\mathtt{OPT}_\pi$ optimizer, which we refer to as $\mathtt{ProST\text{-}T}$ (note that $\mathtt{-T}$ stands for an instance in the tabular setting). We employ the Natural Policy Gradient (NPG) with entropy regularization [17] as a policy update algorithm in $\mathtt{OPT}_\pi$. We denote the entropy regularization coefficient as $\tau$, the learning rate as $\eta$, the policy evaluation approximation gap arising due to finite samples as $\delta$, and the past reference length for forecaster $f$ as $w$. Without loss of generality, we assume that each policy iteration takes one second. The theoretical analysis is conducted within a tabular environment, allowing us to relax Assumption 2, which means that one can estimate non-stationary parameters by counting visitation of state and action pairs at time $t_k$, denoted as $n_{(k)}(s,a)$, rather than observing them. Additionally, we incorporate the exploration bonus term at time $t_k$ into $\widehat{R}_{(k+1)}$, denoted as $\Gamma_w^{(k)}(s,a)$, which is proportional to $\sum_{\tau=k-w+1}^{k}(n_{(\tau)}(s,a))^{-1/2}$ and aims to promote the exploration of states and actions that are visited infrequently.

We compute $\Delta_\pi^*$ by minimizing an upper bound on the $\mathtt{ProST\text{-}T}$'s dynamic regret. The dynamic regret of $\mathtt{ProST\text{-}T}$ is characterized by the *model prediction error* that measures the MDP forecaster's error by defining the difference between $\widehat{\mathcal{M}}_{(k+1)}$ and $\mathcal{M}_{(k+1)}$ through a Bellman equation.

**Definition 3** (Model prediction error). *At time $t_k$, the MDP forecaster predicts a model $\widehat{\mathcal{M}}_{(k+1)}$ and then we obtain a near-optimal policy $\widehat{\pi}^{(k+1)}$ based on $\widehat{\mathcal{M}}_{(k+1)}$. For each pair $(s,a)$, we denote the state value function and the state action value function of $\widehat{\pi}^{(k+1)}$ in $\widehat{\mathcal{M}}_{(k+1)}$ at step $h \in [H]$ as $\widehat{V}_h^{(k+1)}(s)$ and $\widehat{Q}_h^{(k+1)}(s,a)$, respectively. We also denote the model prediction error associated with time $t_{k+1}$ calculated at time $t_k$ as $\iota_h^{(k+1)}(s,a)$, which is defined as*

$$\iota_h^{(k+1)}(s,a) := \left( R_{(k+1)} + \gamma P_{(k+1)} \widehat{V}_{h+1}^{(k+1)} - \widehat{Q}_h^{(k+1)} \right)(s,a).$$

We now derive an upper bound on the $\mathtt{ProST\text{-}T}$ dynamic regret. We expect the upper bound to be likely controlled by two factors: the error of the MDP forecaster's prediction of the future MDP model and the error of the NPG algorithm due to approximating the optimal policy within an estimated future MDP model. This insight is clearly articulated in the next theorem.

**Theorem 1** ($\mathtt{ProST\text{-}T}$ dynamic regret $\mathfrak{R}$). *Let $\iota_H^K = \sum_{k=1}^{K-1} \sum_{h=0}^{H-1} \iota_h^{(k+1)}(s_h^{(k+1)}, a_h^{(k+1)})$ and $\bar{\iota}_\infty^K := \sum_{k=1}^{K-1} \|\bar{\iota}_\infty^{k+1}\|_\infty$, where $\iota_H^K$ is a data-dependent error. For a given $p \in (0,1)$, the dynamic regret of the forecasted policies $\{\widehat{\pi}^{(k+1)}\}_{1:K-1}$ of $\mathtt{ProST\text{-}T}$ is upper bounded with probability at least $1-p/2$ as follows:*

$$\mathfrak{R}\left( \{\widehat{\pi}^{(k+1)}\}_{1:K-1}, K \right) \le \mathfrak{R}_I + \mathfrak{R}_{II}$$

*where $\mathfrak{R}_I = \bar{\iota}_\infty^K/(1-\gamma) - \iota_H^K + C_p \cdot \sqrt{K-1}$, $\mathfrak{R}_{II} = C_{II}[\Delta_\pi] \cdot (K-1)$, and $C_p, C_{II}[\Delta_\pi]$ are some functions of $p, \Delta_\pi$, respectively.*

Specifically, the upper bound is composed of two terms: $\mathfrak{R}_I$ that originates from the MDP forecaster error between $\mathcal{M}_{(k+1)}$ and $\widehat{\mathcal{M}}_{(k+1)}$, and $\mathfrak{R}_{II}$ that arises due to the suboptimality gap between $\pi^{*,(k+1)}$ and $\widehat{\pi}^{(k+1)}$. Theorem 1 clearly demonstrates that a prudent construction of the MDP

forecaster that controls the model prediction errors and the selection of the agent tempo $\Delta_\pi$ is significant in guaranteeing sublinear rates for $\mathfrak{R}_I$ and $\mathfrak{R}_{II}$. To understand the role of the environment tempo in $\mathfrak{R}_I$, we observe that the MDP forecaster utilizes $w$ previous observations, which inherently encapsulates the environment tempo. We expect the model prediction errors, at least in part, to be controlled by the environment tempo $B(\Delta_\pi)$, so that a trade-off between two tempos can be framed as the trade-off between $\mathfrak{R}_I$ and $\mathfrak{R}_{II}$. Hence, it is desirable to somehow minimize the upper bound with respect to $\Delta_\pi$ to obtain a solution, denoted as $\Delta_\pi^*$, which strikes a balance between $\mathfrak{R}_I$ and $\mathfrak{R}_{II}$.

### 4.1.1 Analysis of $\mathfrak{R}_{II}$

A direct analysis of the upper bound $\mathfrak{R}_I + \mathfrak{R}_{II}$ is difficult since its dependence on $K$ is not explicit. To address this issue, we recall that an optimal $\Delta_\pi^*$ should be a natural number that guarantees the sublinearity of both $\mathfrak{R}_I$ and $\mathfrak{R}_{II}$ with respect to the total number of episodes $K$. We first compute a set $\mathbb{N}_{II} \subset \mathbb{N}$ that includes those values of $\Delta_\pi$ that guarantee the sublinearity of $\mathfrak{R}_{II}$, and then compute a set $\mathbb{N}_I \subset \mathbb{N}$ that guarantees the sublinearity of $\mathfrak{R}_I$. Finally, we solve for $\Delta_\pi^*$ in the common set $\mathbb{N}_I \cap \mathbb{N}_{II}$.

**Proposition 1** ($\Delta_\pi$ bounds for sublinear $\mathfrak{R}_{II}$). *A total step $H$ is given by MDP. For a number $\epsilon > 0$ such that $H = \Omega\left(\log\left((\widehat{r}_{max} \vee r_{max})/\epsilon\right)\right)$, we choose $\delta, \tau, \eta$ to satisfy $\delta = \mathcal{O}\left(\epsilon\right)$, $\tau = \Omega\left(\epsilon/\log|\mathcal{A}|\right)$ and $\eta \le (1-\gamma)/\tau$, where $\widehat{r}_{max}$ and $r_{max}$ are the maximum reward of the forecasted model and the maximum reward of the environment, respectively. Define $\mathbb{N}_{II} := \{n \mid n > \frac{1}{\eta\tau}\log\left(\frac{C_1(\gamma+2)}{\epsilon}\right), n \in \mathbb{N}\}$, where $C_1$ is a constant. Then $\mathfrak{R}_{II} \le 4\epsilon(K-1)$ for all $\Delta_\pi \in \mathbb{N}_{II}$.*

As a by-product of Proposition 1, the sublinearity of $\mathfrak{R}_{II}$ can be realized by choosing $\epsilon = \mathcal{O}((K-1)^{\alpha-1})$ for any $\alpha \in [0, 1)$, which suggests that a tighter upper bound on $\mathfrak{R}_{II}$ requires a smaller $\epsilon$ and subsequently a larger $\Delta_\pi \in \mathbb{N}_{II}$. The hyperparameter conditions in Proposition 1 can be found in Lemma 1 and 2 in Appendix D.3.

### 4.1.2 Analysis of $\mathfrak{R}_I$

We now relate $\mathfrak{R}_I$ to the environment tempo $B(\Delta_\pi)$ using the well-established non-stationary adaptation technique of Sliding Window regularized Least-Squares Estimator (SW-LSE) as the MDP forecaster [18–20]. The tractability of the SW-LSE algorithm allows to upper-bound the model predictions errors $\iota_H^K$ and $\bar{\iota}_\infty^K$ by the environment tempo extracted from the past $w$ observed trajectories, leading to a sublinear $\mathfrak{R}_I$ as demonstrated in the following theorem.

**Theorem 2** (Dynamic regret $\mathfrak{R}_I$ with $f =$ SW-LSE). *For a given $p \in (0, 1)$, if the exploration bonus constant $\beta$ and regularization parameter $\lambda$ satisfy $\beta = \Omega(|\mathcal{S}|H\sqrt{\log(H/p)})$ and $\lambda \ge 1$, then $\mathfrak{R}_I$ is bounded with probability at least $1 - p$ as follows:*

$$\mathfrak{R}_I \le C_I[B(\Delta_\pi)] \cdot w + C_k \cdot \sqrt{\frac{1}{w}\log\left(1 + \frac{H}{\lambda}w\right)} + C_p \cdot \sqrt{K-1}$$

*where $C_I[B(\Delta_\pi)] = (1/(1-\gamma) + H) \cdot B_r(\Delta_\pi) + (1 + H\widehat{r}_{max})\gamma/(1-\gamma) \cdot B_p(\Delta_\pi)$, and $C_k$ is a constant on the order of $\mathcal{O}(K)$.*

For a brief sketch of how SW-LSE makes the environment tempo appear in the upper bound, we outline that the model prediction errors are upper-bounded by two forecaster errors, namely $P_{(k+1)}$ - $\widehat{P}_{(k+1)}$ and $R_{(k+1)} - \widehat{R}_{(k+1)}$, along with the visitation count $n_{(k)}(s, a)$. Then, the SW-LSE algorithm provides a solution $(\widehat{P}_{(k+1)}, \widehat{R}_{(k+1)})$ as a closed form of linear combinations of past $w$ estimated values $\{\widetilde{P}, \widetilde{R}\}_{(k-w+1:w)}$. Finally, employing the Cauchy inequality and triangle inequality, we derive two forecasting errors that are upper-bounded by the environment tempo. For the final step before obtaining a suboptimal $\Delta_\pi^*$, we compute $\mathbb{N}_I$ that guarantees the sublinearity of $\mathfrak{R}_I$.

**Proposition 2** ($\Delta_\pi$ bounds for sublinear $\mathfrak{R}_I$). *Denote $B(1)$ as the environment tempo when $\Delta_\pi = 1$, which is a summation over all time steps. Assume that the environment satisfies $B_r(1) + B_p(1)\widehat{r}_{max}/(1-\gamma) = o(K)$ and we choose $w = \mathcal{O}((K-1)^{2/3}/(C_I[B(\Delta_\pi)])^{2/3})$. Define the set $\mathbb{N}_I$ to be $\{n \mid n < K, n \in \mathbb{N}\}$. Then $\mathfrak{R}_I$ is upper-bounded as $\mathfrak{R}_I = \mathcal{O}\left(C_I[B(\Delta_\pi)]^{1/3}(K-1)^{2/3}\sqrt{\log((K-1)/C_I[B(\Delta_\pi)])}\right)$ and also satisfies a sublinear upper bound, provided that $\Delta_\pi \in \mathbb{N}_I$.*

The upper bound on the environment tempo $B(1)$ in proposition 2 is aligned with our expectation that dedicating an excessively long time to a single iteration may not allow for an effective policy approximation, thereby hindering the achievement of a sublinear dynamic regret. Furthermore, our insight that a larger environment tempo prompts the MDP forecaster to consider a shorter past reference length, aiming to mitigate forecasting uncertainty, is consistent with the condition involving $w$ stated in Proposition 2.

### 4.1.3 Suboptimal tempo $\Delta_\pi^*$

So far, we have shown that an upper bound on the `ProST` dynamic regret is composed of two terms $\mathfrak{R}_I$ and $\mathfrak{R}_{II}$, which are characterized by the environment tempo and the agent tempo, respectively. Now, we claim that a suboptimal tempo that minimizes `ProST`'s dynamic regret could be obtained by the optimal solution $\Delta_\pi^* = \arg\min_{\Delta_\pi \in \mathbb{N}_I \cap \mathbb{N}_{II}} (\mathfrak{R}_I^{\max} + \mathfrak{R}_{II}^{\max})$, where $\mathfrak{R}_I^{\max}$ and $\mathfrak{R}_{II}^{\max}$ denote the upper bounds on $\mathfrak{R}_I$ and $\mathfrak{R}_{II}$. We show that $\Delta_\pi^*$ strikes a balance between the environment tempo and the agent tempo since $\mathfrak{R}_I^{\max}$ is a non-decreasing function of $\Delta_\pi$ and $\mathfrak{R}_{II}^{\max}$ is a non-increasing function of $\Delta_\pi$. Theorem 3 shows that the optimal tempo $\Delta_\pi^*$ depends on the environment's drifting constants introduced in Assumption 1.

**Theorem 3** (Suboptimal tempo $\Delta_\pi^*$). *Let* $k_{\textit{Env}} = (\alpha_r \vee \alpha_p)^2 C_I[B(1)]$, $k_{\textit{Agent}} = \log(1/(1-\eta\tau))C_1(K-1)(\gamma+2)$. *Consider three cases: case1:* $\alpha_r \vee \alpha_p = 0$, *case2:* $\alpha_r \vee \alpha_p = 1$, *case3:* $0 < \alpha_r \vee \alpha_p < 1$ *or* $\alpha_r \vee \alpha_p > 1$. *Then* $\Delta_\pi^*$ *depends on the environment's drifting constants as follows:*

- *Case1:* $\Delta_\pi^* = T$.

- *Case2:* $\Delta_\pi^* = \log_{1-\eta\gamma}(k_{\textit{Env}}/k_{\textit{Agent}}) + 1$.

- *Case3:* $\Delta_\pi^* = \exp\left(-W\left[-\frac{\log(1-\eta\tau)}{\max(\alpha_r,\alpha_p)-1}\right]\right)$, *provided that the parameters are chosen so that* $k_{\textit{Agent}} = (1-\eta\tau)k_{\textit{Env}}$.

### 4.2 Improving MDP forecaster

Determining a suboptimal tempo by minimizing an upper bound on $\mathfrak{R}_I + \mathfrak{R}_{II}$ can be improved by using a tighter upper bound. In Proposition 1, we focused on the $Q$ approximation gap $\delta$ to provide a justifiable upper bound on $\mathfrak{R}_I + \mathfrak{R}_{II}$. It is important to note that the factor $\delta$ arises not only from the finite sample trajectories as discussed in [21], but also from the forecasting error between $\mathcal{M}_{(k+1)}$ and $\widehat{\mathcal{M}}_{(k+1)}$. It is clear that the MDP forecaster establishes a lower bound on $\delta$ denoted as $\delta_{\min}$, which in turn sets a lower bound on $\epsilon$ and consequently on $\mathfrak{R}_I$. This inspection highlights that the MDP forecaster serves as a common factor that controls both $\mathfrak{R}_I$ and $\mathfrak{R}_{II}$, and a further investigation to improve the accuracy of the forecaster is necessary for a better bounding on $\mathfrak{R}_I + \mathfrak{R}_{II}$.

Our approach to devising a precise MDP forecaster is that, instead of *selecting* the past reference length $w$ as indicated in Proposition 2, we set $w = k$, implying the utilization of all past observations. However, we address this by solving an additional optimization problem, resulting in a tighter bound on $\mathfrak{R}_I$. We propose a method that adaptively assigns different weights $q \in \mathbb{R}_+^k$ to the previously observed non-stationary parameters up to time $t_k$, which reduces the burden of choosing $w$. Hence, we further analyze $\mathfrak{R}_I$ through the utilization of the Weighted regularized Least-Squares Estimator (`W-LSE`) [22]. Unlike `SW-LSE`, `W-LSE` does not necessitate a predefined selection of $w$, but it instead engages in a joint optimization procedure involving the data weights $q$ and the future model $(\widehat{P}_{(k+1)}, \widehat{R}_{(k+1)})$. To this end, we define the forecasting reward model error as $\Delta_k^r(s,a) = |(R_{(k+1)} - \widehat{R}_{(k+1)})(s,a)|$ and the forecasting transition probability model error as $\Delta_k^p(s,a) = \|(P_{(k+1)} - \widehat{P}_{(k+1)})(\cdot \mid s,a)\|_1$.

**Theorem 4** ($\mathfrak{R}_I$ upper bound with $f$=`W-LSE`). *By setting the exploration bonus* $\Gamma_{(k)}(s,a) = \frac{1}{2}\Delta_k^r(s,a) + \frac{\gamma\tilde{r}_{max}}{2(1-\gamma)}\Delta_k^p(s,a)$, *it holds that*

$$\mathfrak{R}_I \le \left(4H + \frac{2\gamma|\mathcal{S}|}{1-\gamma}\left(\frac{1}{1-\gamma} + H\right)\right)\left(\frac{1}{2}\sum_{k=1}^{K-1}\Delta_k^r(s,a) + \frac{\gamma\tilde{r}_{max}}{2(1-\gamma)}\sum_{k=1}^{K-1}\Delta_k^p(s,a)\right).$$

**Remark 1** (Tighter $\mathfrak{R}_\mathcal{I}$ upper bound with $f$ = W-LSE)**.** *If the optimization problem of W-LSE is feasible, then the optimal data weight $q^*$ provides tighter bounds for $\Delta_k^r$ and $\Delta_k^p$ in comparison to SW-LSE, consequently leading to a tighter upper bound for $\mathfrak{R}_\mathcal{I}$. We prove in Lemmas 4 and 6 in Appendix D.3 that $\bar{\iota}_\infty^K$ and $-\iota_H^K$ are upper-bounded in terms of $\Delta_k^r$ and $\Delta_k^p$.*

### 4.3 ProST-G

The theoretical analysis outlined above serves as a motivation to empirically investigate two key points: firstly, the existence of an optimal training time; secondly, the role of the MDP forecaster's contribution to the ProST framework's overall performance. To address these questions, we propose a practical instance, named ProST-G, which particularly extends the investigation in Section 4.2. ProST-G optimizes a policy with the soft actor-critic (SAC) algorithm [23], utilizes the integrated autoregressive integrated moving average (ARIMA) model for the proactive forecaster $f$, and uses a bootstrap ensemble of dynamic models where each model is a probabilistic neural network for the model predictor $g$. We further discuss specific details of ProST-G in Appendix F.3 and in Algorithm 3.

## 5 Experiments

We evaluate ProST-G with four baselines in three Mujoco environments each with five different non-stationary speeds and two non-stationary datasets.

**(1) Environments: Non-stationary desired posture.** We make the rewards in the three environments non-stationary by altering the agent's desired directions. The forward reward $R_t^f$ changes as $R_t^f = o_t \cdot \bar{R}_t^f$, where $\bar{R}_f$ is the original reward from the Mujoco environment. The non-stationary parameter $o_k$ is generated from the sine function with five different speeds and from the real data $A$ and $B$. We then measure the time-elapsing variation budget by $\sum_{k=1}^{K-1} |o_{k+1} - o_k|$. Further details of the environment settings can be found in Appendix D.1.1.

**(2) Benchmark methods.** Four baselines are chosen to empirically support our second question: the significance of the forecaster. **MBPO** is the state-of-the-art model-based policy optimization [24]. **Pro-OLS** is a policy gradient algorithm that predicts the future performance and optimizes the predicted performance of the future episode [7]. **ONPG** is an adaptive algorithm that performs a purely online optimization by fine-tuning the existing policy using only the trajectory observed online [8]. **FTRL** is an adaptive algorithm that performs follow-the-regularized-leader optimization by maximizing the performance on all previous trajectories [9].

## 6 Discussions

### 6.1 Performance compare

The outcomes of the experimental results are presented in Table 1. The table summarizes the average return over the last 10 episodes during the training procedure. We have illustrated the complete training results in Appendix E.3. In most cases, ProST-G outperforms MBPO in terms of rewards, highlighting the adaptability of the ProST framework to dynamic environments. Furthermore, except for data $A$ and $B$, ProST-G consistently outperforms the other three baselines. This supports our motivation of using the proactive model-based method for a higher adaptability in non-stationary environments compared to state-of-the-art model-free algorithms (Pro-OLS, ONPG, FTRL). We elaborate on the training details in Appendix E.2.

Table 1: Average reward returns

| Speed | $B(G)$ | Swimmer-v2 | | | | | Halfcheetah-v2 | | | | | Hopper-v2 | | | | |
|---|---|---|---|---|---|---|---|---|---|---|---|---|---|---|---|---|
| | | Pro-OLS | ONPG | FTML | MBPO | ProST-G | Pro-OLS | ONPG | FTML | MBPO | ProST-G | Pro-OLS | ONPG | FTML | MBPO | ProST-G |
| 1 | 16.14 | -0.40 | -0.26 | -0.08 | -0.08 | **0.57** | -83.79 | -85.33 | -85.17 | -24.89 | **-19.69** | **98.38** | 95.39 | 97.18 | 92.88 | 92.77 |
| 2 | 32.15 | 0.20 | -0.12 | 0.14 | -0.01 | **1.04** | -83.79 | -85.63 | -86.46 | -22.19 | **-20.21** | 98.78 | 97.34 | **99.02** | 96.55 | 98.13 |
| 3 | 47.86 | -0.13 | 0.05 | -0.15 | -0.64 | **1.52** | -83.27 | -85.97 | -86.26 | -21.65 | **-21.04** | 97.70 | 98.18 | 98.60 | 95.08 | **100.42** |
| 4 | 63.14 | -0.22 | -0.09 | -0.11 | -0.04 | **2.01** | -82.92 | -84.37 | -85.11 | -21.40 | **-19.55** | 98.89 | 97.43 | 97.94 | 97.86 | **100.68** |
| 5 | 77.88 | -0.23 | -0.42 | -0.27 | 0.10 | **2.81** | -84.73 | -85.42 | -87.02 | **-20.50** | -20.52 | 97.63 | 99.64 | 99.40 | 96.86 | **102.48** |
| A | 8.34 | 1.46 | 2.10 | **2.37** | -0.08 | 0.57 | -76.67 | -85.38 | -83.83 | -40.67 | **83.74** | 104.72 | **118.97** | 115.21 | 100.29 | 111.36 |
| B | 4.68 | **1.79** | -0.72 | -1.20 | 0.19 | 0.20 | -80.46 | -86.96 | -85.59 | -29.28 | **76.56** | 80.83 | **131.23** | 110.09 | 100.29 | 127.74 |

## 6.2 Ablation study

An ablation study was conducted on the two aforementioned questions. The following results support our inspection of Section 4.2 and provide strong grounds for Theorem 3.

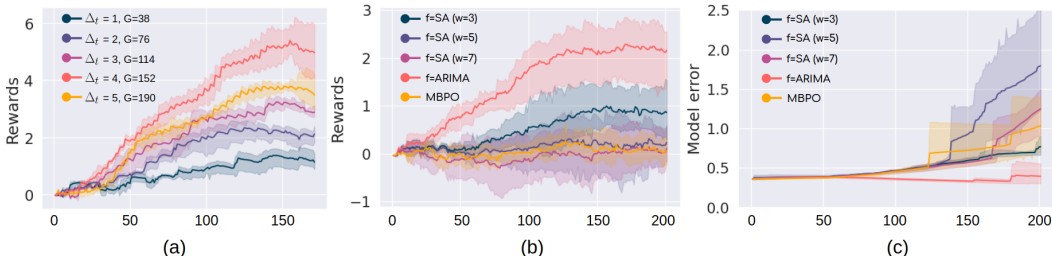

Figure 3: (a) Optimal $\Delta_\pi^*$; (b) Different forecaster $f$ (ARIMA, SA); (c) The Mean squared Error (MSE) model loss of `ProST-G` with four different forecasters (ARIMA and three SA) and the MBPO. The $x$-axis in each figure shows the episodes.

**Suboptimal $\Delta_\pi^*$.** The experiments are performed over five different policy training times $\Delta_\pi \in \{1, 2, 3, 4, 5\}$, aligned with SAC's number of gradient steps $G \in \{38, 76, 114, 152, 190\}$, under a fixed environment speed. Different from our theoretical analysis, we set $\Delta_t = 1$ with $G = 38$. We generate $o_k = sin(2\pi\Delta_\pi k/37)$, which satisfies Assumption 1 (see Appendix E.1). The shaded areas of Figures 3 (a), (b) and (c) are 95 % confidence area among three different noise bounds of 0.01,0.02 and 0.03 in $o_k$. Figure 3(a) shows $\Delta_t = 4$ is close to the optimal $G^*$ among five different choices.

**Functions $f, g$.** We investigate the effect of the forecaster $f$'s accuracy on the framework using two distinct functions: ARIMA and a simple average (SA) model, each tested with three different the values of $w$. Figure 3(b) shows the average rewards of the SA model with $w \in \{3, 5, 7\}$ and ARIMA model (four solid lines). The shaded area is 95 % the confidence area among 4 different speeds $\{1, 2, 3, 4\}$. Figure 3(c) shows the corresponding model error. Also, we investigate the effect of the different model predictor $g$ by comparing MBPO (reactive-model) and `ProST-G` with $f$ =ARIMA (proactive-model) in Figure 3(c). The high returns from `ProST-G` with $f$ = ARIMA, compared to those from MBPO, empirically support that the forecasting component of the **ProST** framework can provide a satisfactory adaptability to the baseline algorithm that is equipped with. Also, Figures 3(b) and 3(c) provide empirical evidence that the accuracy of $f$ is contingent on the sliding window size, thereby impacting the model accuracy and subsequently influencing the agent's performance.

## 7 Conclusion

This work offers the first study on the important issue of time synchronization for non-stationary RL. To this end, we introduce the concept of the tempo of adaptation in a non-stationary RL, and obtain a suboptimal training time. We propose a Proactively Synchronizing Tempo (ProST) framework, together with two specific instances `ProST-T` and `ProST-G`. The proposed method adjusts an agent's tempo to match the tempo of the environment to handle non-stationarity through both theoretical analysis and empirical evidence. The ProST framework provides a new avenue to implement reinforcement learning in the real world by incorporating the concept of adaptation tempo.

As a future work, it is important to generalize the proposed framework to learn a safe guarantee policy in a non-stationary RL by considering the adaptation tempo of constraint violations [25, 26]. Another generalization is to introduce an alternative dynamic regret metric, enabling a fair performance comparison among agents, even when they have varying numbers of total episodes. Another future work is to find an optimal tempo of the distribution correction in offline non-stationary RL, specifically how to adjust the relabeling function to offline data in a time-varying environment that is dependent on the tempo of the environment [27, 28].

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
