# OpenReview forum: "Tempo Adaptation in Non-stationary Reinforcement Learning"
_NeurIPS.cc/2023/Conference — NeurIPS 2023 poster_

### Official Review · Reviewer_3oD2 · 2023-07-02

**Soundness:** 3 good
**Presentation:** 2 fair
**Contribution:** 3 good
**Rating:** 5
**Confidence:** 3

**Summary:**

This paper studies reinforcement learning in non-stationary environments. The authors propose a proactive tempo-control model-based (PTM) framework to adjust how often the learning agent to adjust its policy to the environment tempo. Two different variants of PTM are considered: The PTM-T variant uses natural policy gradient as the base algorithm and achieves sublinear dynamic regret. The regret bound is further improved when the forecaster assigns different weights to the observed data. The second variant PTM-G is more empirical and is shown to outperform four existing baselines on three different Mujoco tasks.

**Strengths:**

This paper studies an interesting and important problem in reinforcement learning: The quick adaptation of the learning algorithm to a non-stationary environment. The proposed solution of proactive tempo-control is interesting and novel as far as I know. The theoretical derivations look solid and convincing. This paper also includes a decent amount of numerical results with ablation studies in addition to the theoretical contributions, which also looks encouraging.

**Weaknesses:**

1.	While I am sure that the authors have a lot of interesting results to share, the writing of the paper fails to convey some of the important ideas, which prevents me from fully appreciating the contributions of the work. The presentations of the methodology and results in the paper are somewhat unclear and sometimes can be confusing. There are a few things that are unclear to me and can certainly be improved:

a. The main theoretical results are not clearly stated. The authors build up the theoretical results through a series of theorems and propositions, but it is hard to keep track of what the strongest results are, given that there are numerous results presented. The reader needs to combine a few theorems to see the exact form of the dynamic regret bound, which is not reader-friendly. The authors could consider providing the strongest, take-home results of the work in one theorem.

b. The main text of the paper is not very self-contained, which adds to the confusion. The definitions of some notations and even algorithm descriptions (e.g., Algorithms 1 and 2) are deferred to appendices, which really should have been included in the main text. My suggestion is that some of the intermediate results in Section 4 can be moved to the appendices to allow more space for many other sections.

2.	The main setup of the problem formulation is a bit unclear to me. The authors mentioned the main “trade-off” between interacting with the environment more often to gather data vs. performing more policy optimization using collected samples. Why is it even a trade-off and why can’t we do both of them at the same time? The only reason that I can think of is computation complexity, which is less critical and was not even discussed in the paper.

3.	It is unclear to me how the problem formulation is related to the standard setup in non-stationary RL, e.g., [Cheung et al., 2020]. The definition of the time elapsing variation budget is a bit ad hoc and should be better explained. It is also quite confusing that the formulation considers both a finite-horizon $H$ and a reward discounting factor $\gamma$, while usually people only choose one of the two. The definitions of the values functions are also missing (which are included in the appendix but should definitely have been included in the main text).

4.	The assumption that $B_p(\Delta_t) = c^{\alpha_p} B_p(\Delta_t)$ seems a bit ad hoc to me and probably needs more justification. Similarly, it is also unclear what Assumption 1 means and the authors could provide some examples on how it is satisfied.

5.	In the simulations, it seems that some of the selected comparison baselines are originally designed for the stationary environment. It would be unfair to compare with these algorithms in non-stationary environments. I wonder how the proposed method compares to some existing solutions specifically designed for non-stationary RL (e.g., some simple sliding-window or restarting-based algorithms).

6.	A related work section is missing (and again is included in the appendix but should have been included in the main text to be self-contained). There are also a few important closely related works that are not discussed in the references. Just to name a few that I am aware of:

a.	Wei, Chen-Yu, and Haipeng Luo. "Non-stationary reinforcement learning without prior knowledge: An optimal black-box approach." Conference on learning theory, 2021.

b.	Mao, Weichao, et al. "Near-optimal model-free reinforcement learning in non-stationary episodic mdps." International Conference on Machine Learning, 2021.

**Questions:**

Please consider first answering my questions in the “Weaknesses” section. The additional questions listed below are comparably not major.

1.	How does the environment tempo formulation deal with the difference between the frequency of environment non-stationarity (i.e., frequent, slow, gradual variations) and the magnitude of non-stationarity (i.e., abrupt variations).

2.	In Eq (2.1), the authors measure the transition variations in terms of the infinity-norm, which is different from the more popular metric of 1-norm in existing non-stationary RL works (e.g., Cheung et al., 2020). I wonder why such a difference occurs.

3.	Since the authors mentioned the word “meta-algorithm”, I wonder how the proposed method is related to meta-learning?

**Limitations:**

The limitations are briefly discussed in the conclusions section.

---

> ### Author Rebuttal · Authors · 2023-08-08
>
> $\textbf{1)  W 1,2: clairty of main contribution and meaning of tradeoff}$
>
> Thanks for pointing out the clarity of the main theoretical results and corresponding contribution and the meaning of the “trade-off”.  We would like to emphasize our work’s main contribution is interpreting the non-stationary MDP as “$\textit{wall-clock time}$” goes by, not as “$\textit{episode}$” goes by, and shifting the perspective from episode to time is a more realistic setting that matches with non-stationary RL’s motivation: RL for real-world application. Raising this issue recalls our new problem setting stated in the introduction (line 2 $\sim$ 5). This new setting gives rise to solving an additional problem: how to choose the optimal $K-$length time sequence that the agent interacts with the environment?”. With this respect, our main theoretical result is that we determine the optimal K-lenght time sequence by optimal training time, and we propose a PTM framework to find optimal training time.
>
> Now, we would like to address the “tradeoff”. Yes, the tradeoff exists between the agent’s training time and how fast the environment changes, but it's not for the fixed episode(K), but for fixed “time duration”. For better understanding, we would like to remind the example of an introduction with a detailed explanation. Let’s say for a fixed time duration 0[s]$\sim$15[s], robot A executes episodes for every 1 second, [0,1,..,15], and between time $t,t+1$, the agent trains the policy. Robot B executes episodes every 2 seconds [0,2,..,14] and between time $t,t+2$, the agent trains the policy. Let’s assume one policy update takes 1 second. Then, the robot A can update one time, and the robot B can update two times. The robot A interacts with the environment for 15 episodes and updates policy one time, which has more information about the environment but has an uncertain approximate optimal policy. Robot B interacts with the environment for 8 episodes and updates policy two times, which has less information about the environment but has a better approximate optimal policy than robot A.
>
> The above results clearly show the existence of tradeoff (line53~54) and support our main contribution that how to choose optimal training time is significant that should be addressed.
>
> $\textbf{2) W3: necessity of defining time elasping variation budget}$
>
> Thanks for pointing out the necessity of defining time elapsing variation budget compared to the standard variation budget [Cheung et al. 2020]. Aside from the standard definition of variation budget, we need a new definition since the time of the environment and the episode of the agent are not synchronized anymore. Standard setup [Cheung et al. 2020] defines the variation budget as the difference between two consecutive MDP and takes summation over the total K episodes. However, in our setting, how the agent chooses the interaction time $[t_1,t_2,...,t_K]$ brings about different variation budgets even though executing total K episodes (K interaction time instances) since the environment changes as time goes by. Therefore, two different agents that execute with the same K episodes, but different interaction time sequence experience different variation budgets. This is the reason why we need to define variation budget with respect to the environment’s time, which we named “time-elapsing variation budget”.
>
> $\textbf{3) W4: Necessity of the property$B_p(\Delta_t) = c^{\alpha_p} B_p(\Delta_t)$}$
>
> The main reason why we added this assumption is to justify our new definition, time-elapsing variation budget, covers and can represent standard MDPs, so utilizing our practical setting can be compatible with previous non-stationary RL methods. (line 118 $\sim$ line121). $\alpha_p = \alpha_r =0$ represents a stationary environment (typo on the main paper line 120 $\sim$ 121) and $\alpha_p = \alpha_r=1$ is a linear drifiting environment. This validness of our assumption is well-supported by the result of our main theorem, Theorem 3. In theorem 3, as case1: $max(\alpha_r, \alpha_p) =0$ , the $G^*$ is infinity. This makes sense, since in a stationary environment, a large number of NPG gradient steps (iteration numbers of policy updates) guarantee policy closer to the optimal policy.
>
> $\textbf{4) W5: appropriateness of the baseline algorithms}$
>
> We admit MBPO algorithm is built upon a stationary environment. The main reason why we added the MBPO algorithm is to compare the result between MBPO and PTM-G framework. (line 335 $\sim$ 338). Note that the PTM-G framework is a model-based policy optimization framework, but the big deviation from MBPO occurs PTM-G framework predicts the future non-stationary model using the MDP forecaster and the MBPO predicts the stationary-model by past observed trajectories. This structure difference clearly shows comparing MBPO and PTM-G framework can shed light on the forecaster performance of the PTM-G framework. We also have selected left three algorithms as a baseline, since those are algorithms that were proposed to deal with non-stationarity and have been utilized as a baseline algorithm [13].
>
> $\textbf{5) Q1}$ : The environment tempo, which is a time-elapsing variation budget, can capture both properties of the non-stationarity since it’s s a definition that captures the non-stationarity with respect to time, not respect to episode.
>
> $\textbf{6) Q2}$ : We apologize for the typo. The variations of the transition probability should $\sum_{k=1}^{K-1} \sup_{s,a} || P^{t_{k+1}} ( \cdot | s,a) - P^{t_{k}} ( \cdot | s,a) ||$.
>
> $\textbf{ 7) Q3}$ : We first note that our PTM framework is not related to meta-learning. We use the term “meta-algorithm” since the PTM framework 1) first provides an MDP forecaster that creates a future MDP 2) then obtains optimal policy in that model.  The term “meta-algorithm” represents we can use any existing RL algorithms that obtain an optimal policy for a given model in process 2) (line120 $\sim$ line 151).

---

> > ### Comment · Reviewer_3oD2 · 2023-08-19
> >
> > I thank the authors for the responses, which helped address some of my concerns. I am still conservative about whether the time synchronization issue is an important one, but I am happy to increase my score to the positive side.

---

> ### Author Response · Authors · 2023-08-16
> **Looking for the feedback!**
>
> Dear Reviewer 3oD2,
>
> We first thank you for the positive feedback on our work's results provided in the initial official review. Since the discussion deadline is approaching, we wonder whether our above response to your initial review was fully addressed to your concerns. We appreciate your comments that a low rating mainly comes from the unclarity of the paper wiring, especially theoretical analysis, and the appropriateness of the baselines in experiments. We believe that we have addressed your doubts and concerns clearly in our rebuttal. Particularly, we also highlight our work's main contribution and motivation for you to accelerate resolving the above concerns.
>
> __We wonder whether our comments are fully addressed to you, or if not, we would love to hear back from you!__
>
> Best regards,
>
> Authors

---

### Official Review · Reviewer_WHPE · 2023-07-04

**Soundness:** 3 good
**Presentation:** 2 fair
**Contribution:** 2 fair
**Rating:** 4
**Confidence:** 5

**Summary:**

The paper introduces a novel framework for handling non-stationary environments in reinforcement learning (RL) called Proactive Tempo-control Model-based (PTM) framework. The authors argue that in non-stationary environments, an additional factor emerges alongside the classical exploration-exploitation trade-off: the tempo of adaptation. The PTM framework is designed to adjust the policy-update tempo (algorithm tempo) to match the environment tempo. The authors propose two instances of the PTM framework, PTM-T and PTM-G, and provide theoretical analysis and empirical evidence to support their claims.

**Strengths:**

- The concept of tempo adaptation in non-stationary RL environments is an interesting approach.
- The proposed PTM framework could have implications for the field of RL, particularly in real-world applications where environments are often non-stationary.
- The paper provides theoretical analysis for the proposed method.

**Weaknesses:**

- Although the author tells an interesting story about "adapt to adapt," the proposed PTM algorithm essentially attempts to learn a gradually changing MDP, which does not seem to have any fundamental difference from some previous works like [1][2].
- There is a significant gap between the theoretical results of the paper and the practical methods. Although the author has used a considerable amount of theoretical analysis in the main text to try to demonstrate that PTM method can achieve smaller dynamic regret with sublinear iterations, only simple performance curves are shown in the experimental section without further demonstrating the relationship between dynamic regret and iterations, which should explain the principle of improvement in the PTM algorithm.
- As cited and discussed by the author (line 29-30), some previous works have also been able to effectively address non-stationarity. However, in the selected baseline, MBPO is just a standard model-based RL algorithm without specific optimization for non-stationarity. Additionally, other baselines seem to be relatively outdated. Why not compare with newer references like [3], etc?
- This paper uses a lot of complex and scattered math symbols, making it confusing for readers when reading the theoretical part. For example, in line 207, the author uses an icon symbol to represent "Alg". In line 264, the author uses diamond and square symbols to represent $r$, $p$ and $\mathcal{P}$, etc. Why not simply use the letters or with some subscriptions directly? In some contextual formulas (e.g. formula 4.2, 4.3, 4.4, 4.6 and line 271-276), the author interchangeably uses these icon symbols and original letters. When readers read through this paper, they may need to repeatedly recall what these complex symbols actually represent.

[1] Deep reinforcement learning amidst continual structured non-stationarity. Xie, et al. (ICML 2021)

[2] Meta-reinforcement learning by tracking task non-stationarity. Poiani, et al. (IJCAI 2021)

[3]Factored adaptation for non-stationary reinforcement learning. Feng, et al. (NeurIPS 2022)

**Questions:**

- In proposition 3, how are the different cases of environment tempos reflected in the experimental setting?
- In Figure b-2, the learning curves of the algorithms seem to be in their early stages. Why not plot the complete curve until convergence?

**Limitations:**

- The writing and logic of this paper is not good enough, making it hard for readers to follow.
- There is a gap between theory and practice, and there are no detailed experimental demonstrations for theoretical conclusions.
- Additionally, there is a lack of experiments comparing with the recent SOTA baselines.
- Although the author proposes an interesting perspective on "adapt to adapt," fundamentally it does not differ from works on actively learning changes in MDPs.

I suggest that the author reorganize the use of math symbols and consider simplifying unnecessary theoretical analysis in the main text. Furthermore, provide a more detailed explanation of the PTM-G algorithm in Section 4.2 and supplement with additional experiments as mentioned above.

---

> ### Author Rebuttal · Authors · 2023-08-08
>
> We appreciate your constructive comments and the followings are our answers to weakness, question that the reviewer has raised.
>
> $\textbf{1) W1: clarity of contribution and difference from related works}$
>
> First, We would like to emphasize that our work’s $\textbf{main contribution}$ is interpreting the non-stationary MDP as “wall-clock time” goes by, not as “episode” goes by, and shifting the perspective from episode to time is a more realistic setting that matches with non-stationary RL(NSRL)’s motivation: RL for real-world application. Raising this issue recalls our new problem setting stated in the introduction (line 2 $\sim$ 5). This new setting gives rise to solving an additional problem: how to choose the optimal $K-$length time sequence that the agent interacts with the environment?”. With this respect, our $\textbf{main theoretical result}$ is that we determine the optimal K-length interaction time sequence by finding an optimal training time of the agent. Within this sense, we propose the PTM framework to 1) find optimal training time by forecasting model-based method and 2) claim our method yields a better online return, which 1) is supported by Fig 3-(a) and 2) is supported by Table 1.
>
> Therefore, we claim  $\textbf{we unearth a hidden assumption}$ that the agent and the environment should have independent timelines, and our method finds how to synchronize two timelines theoretically with empirical results, which supports we have 1) different problem-setting and 2) different purposes of the algorithm aside from existing the non-stationary RL methods.
>
> Also, please note that the fundamental goal of NSRL methods is to learn gradually changing MDP [p1]. So if your understanding of our proposed method well matches with learning a gradually changing MDP, then it means the proposed method has solved the standard NSRL goal, which support the credibility of our work.
>
> [p1] Milano et al, 2021, State of the Art on: Non-Stationary Reinforcement Learning
>
> $\textbf{2) W2: gap between theoretical results and a practical method}$
>
> We would like to provide two points that how theoretical results (PTM-T) are closely related to the practical algorithm (PTM-G).
>
> Before, we would like to recommend interpreting theoretical analysis not as “theoretical justification or theoretical proof” of the practical algorithm, but rather as “theoretical grounds or insights”, and experiment on ablation study (subsection [6.2]) supports this. Our theoretical analysis can be grouped into two main contributions: 1) the subsection[4.1.1] highlights the existence of optimal tempo, 2) subsection [4.1.2] highlights improving forecaster accuracy is important in PTM framework.
>
> First, the motivation of theoretical analysis on “the existence of optimal tempo” is to support our new problem-setting: additional factors, tempo, arise in a non-stationary environment, and how to adapt the policy-update tempo to the environment tempo is significant. (line 55 $\sim$ line 61). This “optimal tempo existence” is supported by the experiment result in Figure 3-(a) and line 323 $\sim$ line 329.
>
> Also, the motivation of theoretical analysis on “improving forecaster accuracy” is to emphasize “MDP forecaster” serves a significant role in obtaining accurate optimal interaction time sequence. To be specific, finding the optimal tempo by taking minimum of the upperbound of $\mathfrak{R}_\mathcal{I} + \mathfrak{R}_A$ still leave room for an improvement that tighter upperbound provides a much more accurate optimal tempo.
>
> In proposition 1, let's focus on the factor $\delta$ to provide a compact representation of the dynamic regret of PTM-T (Theorem 1). We emphasize that the approximation gap $\delta$ emerges not only due to finite sample trajectories (Lemma 3) , but also the forecasting error between $\mathcal{M}^{k+1}$ and $\widehat{\mathcal{M}}^{k+1}$. It's straightforward since MDP forecaster yields a lowerbound of $\delta$ as $\delta_{\text{min}}$ and $\delta_{\text{min}}$ provides a lowerbound on $\epsilon$ and subsequently a lowerbound on $\mathfrak{R}_\mathcal{I}$. This shows MDP forecaster serves as a common factor that controls both $\mathfrak{R}_I$ and $\mathfrak{R}_{A}$.
>
> Second, the experiment results of PTM-T which exactly match the setting of theoretical analysis are stated in the introduction (in Figure 1-(c)) to utilize its results for motivation.
>
> $\textbf{3) W3 : appropriateness of baseline algorithms}$
>
> First, we would like to claim that we used MBPO as baseline to compare the result between MBPO and PTM-G. (line 335 $\sim$ 338). PTM-G is a model-based policy optimization framework. The difference is that MBPO predicts stationary model and PTM-G predicts the non-stationary model by MDP forecaster. This structure difference clearly supports that comparing MBPO and PTM-G can highlight the forecaster performance of the PTM-G. Please note that the role of the forecaster is significant as we mentioned in section [4.1.2], and we have come up with theoretical soundness (Theorem3, Remark1) and experimental soundness (Figure 3-(b-1),(b-2)) to support its significance.
>
> Second, We provide $\textbf{why} we have selected the three baseline algorithms: ProOLS, FTML, ONPG. We claim those are appropriate baselines to compare the effectiveness of PTM-G. To be specific, ProOLS predicts future policy evaluation based on past observation data but it is a model-free method. ProOLS and PTM-G both shares the “forecasting” component but ProOLS is a model-free method and PTM-G  is a model-based method. Comparing ProOLS and PTM-G can highlight the effect of the "model-based” approach of PTM-G. Other two were also baselines in the non-stationary RL works [13]
>
> $\textbf{4) Q1}$ : The environment tempo's corresponding scenarios are in line 120 $\sim$ 121.  $\alpha_r=\alpha_p=0$ is a stationary environment (typo on main paper)
>
> $\textbf{5) Q2}$ : we rolled 200 episodes. The model error except f=ARIMA increases as episode goes by.

---

> > ### Comment · Reviewer_WHPE · 2023-08-14
> >
> > Thanks for the author's response. Some of my concerns have been addressed. However, I still believe that there are some shortcomings in this work:
> >
> > - Although the authors attempt to explain the theoretical analysis as "theoretical grounds or insights" in their response, such a lengthy and hard-to-follow theoretical analysis is only used to introduce relatively simple algorithm designs as insights. The actual methods and theoretical analysis do not seem to align well. Additionally, as I mentioned in my review, all experimental results in the paper focus on performance. These indirect results do not effectively demonstrate the rationality and necessity of each component in method design and theoretical analysis. Therefore, I still think it is necessary to add separate indicators specifically for these theoretical claims.
> >
> > - The authors do not respond to my Weakness 4. I still believe that improvements can be made regarding complex symbol usage in the paper. Furthermore, the author should consider reorganizing the writing by moving some theoretical analysis to the appendix while retaining only the most important parts and adding clearer logical structure. This will help readers better grasp key insights into method design.
> >
> > For now, I will maintain my current score and continue paying attention to other reviewers' opinions and discussions.

---

> > > ### Author Response · Authors · 2023-08-14
> > > **Correction on misunderstanding and thanks for the feedbacks.**
> > >
> > > Thanks for the constructive feedbacks again and we appreciate the weakness of our work that you raised. We also thank raising additional concerns that were not fully addressed to the Reviewer WHPE.
> > >
> > > The followings are comments that __we want to correct some misunderstandings__ based on the Reviewer WHPE's official comment.
> > > We also welcome further discussions if there exist concerns to be fully clarified.
> > >
> > > ---
> > > $\textbf{[Bullet 1] Regarding with (1) All experiments are focused on the \textcolor{purple}{performance} (2) So, experiments do not match with \textcolor{teal}{each component in the method design}}$ $\textbf{ (3) This makes the critical gap between the theoretical results and experiments}$
> > >
> > > Thanks for re-raising this issue.
> > > We admit that  __misunderstanding__ of above (1), (2) can lead to (3), which makes the reader understand the gap between theorems and experiments can be large.
> > > So we want to __correct__ it as follows.
> > >
> > > First, all experiments are not focused on the performance between PTM-G and the four baselines.
> > >
> > > To be specific,
> > >
> > > *  __Table1__ in __section [6.1. Performance compare] /  Figure 5 $\sim$13__ in __Appendix__ : Focused on the __performance__. This is the numerical results of the average return over the last 10 episodes over 200 episodes between PMT-G and four baselines. Figure 5 $\sim$13 in Appendix shows the whole traning result, where Table 1 is the summary result of the whole training result.
> > > * __Figure 3__  in __section [6.2 Ablation study]__ : Focused on the importance of __each components in method design__ .
> > >
> > >    - Figure 3-(a)  : Focused on the existence of optimal training time $G^* \leftarrow$ supported by Proposition 3 / Section 4.1.1
> > >
> > >    - Figure 3-(b-1),(b-2): Focused on the importance of MDP forecaster design $\leftarrow$ supported by Theorem 3 /Section 4.1.2
> > >
> > > We also mentioned the above points in the above rebuttal as the answer of $\textbf{W2}$, but understood our answer was not fully supportive.
> > > We believe the above bullets are helpful for the reviewer to understand the structure of the paper and take the first step to solve this concern (especially, how experiments are related to theorems / which experiment results show performance and which experiment results show the importance of component)
> > >
> > > __We kindly recommend: 1) please take a look above bullet 2) then please go back to our rebuttal comments on [(2) W2: the gap between theoretical results and a practical method].__
> > >
> > > We would love to hear back from the Reviewer WHPE if this concern is fully addressed, or if not, we welcome some further discussions.
> > >
> > >
> > >
> > > ---
> > >
> > > $\textbf{[Bullet 1] Regarding with (1) Understanding the theoretical analysis as ``theoretical grounds or insights"}$ $\textbf{ (2) But complex and hard-to-follow writings on theoretical analysis make it hard to understand.}$
> > >
> > > Thanks for raising this writing issue on theoretical analysis. We appreciate that the reader should take the burden to fully understand, and we also admit it's hard to follow at first glance. We will organize the theoretical analysis with fewer notations and make it much more clear in the camera-ready version if we get accepted.
> > >
> > >
> > > ---
> > > $\textbf{[Bullet 2] Regarding with paper writing issue}$
> > >
> > >
> > > We admit the paper is not reader-friendly, so the reader has to take the burden to collect the theorems and experiments to understand our main contributions.
> > > We would love to say we will organize the paper in a much more straightforward and make it clear (as you suggest, make theorems shorter and straightforward, use fewer notations to make the paper much more readable) in a camera-ready version if we get accepted. (We omit to address your weakness 4 and want to apologize. The same issue was also raised by the reviewer EXsm and we have answered it. )

---

> ### Author Response · Authors · 2023-08-20
> **Kindly request your confirmation on a misunderstanding.**
>
> Dear Reviewer WHPE,
>
> We sincerely appreciate your active involvement in clarifying the misunderstandings. The comprehensive feedback provided has greatly facilitated to resolve your concerns at hand. __We kindly request your confirmation whether the aforementioned concern outlined in bullet point 1 was resolved. We believe it's a misunderstanding__ that actually we have both 1) experiments on performance between baselines and our proposed algorithm, and 2) experiments to support our algorithm's component design in the main paper. __We have clearly addressed it as above__ : $\textbf{[Bullet1] : Regarding with (1) All experiments are }\sim$. We also acknowledge that the $\textbf{other two bullet points}$ are commonly associated with the issue in paper writing, namely, a lack of reader-friendliness and the inclusion of intricate mathematical notations that require readers to gather multiple theorems in order to comprehend our contribution. If the paper gets accepted, we are willing to refine the presentation of mathematical notations, aiming to render them more succinct, while concurrently simplifying the theorems to enhance their accessibility in the final version.
>
> Best regards,
>
> Authors of the paper 8588

---

> ### Author Response · Authors · 2023-08-21
> **Kindly reminder that we do have experiments in the paper that the reviewer raise in comments.**
>
> Dear Reviewer WHPE,
>
>
>
>
> We genuinely appreciate your effort and detailed feedback to improve the paper. Since this is the last minute of the discussion period, we still believe that __we__ $\textbf{do have experiments}$ __in the main paper__ $\textbf{Table1, Figure3}$ __and in the appendix__ $\textbf{Figure5} \sim \textbf{13}$ __that the reviewer was concerned in bullet 1__. This is a kindly reminder the aforementioned concern outlined in bullet point 1 was addressed to the reviewer. We would like to hear back whether regarding misunderstanding was solved. Thanks!
>
> Best regards,
>
> Authors of the paper 8588

---

> > ### Comment · Reviewer_WHPE · 2023-08-22
> >
> > I appreciate the author's response, but as mentioned earlier, despite my concerns being partially addressed, I still believe that this paper has not fully met the standards for acceptance. Therefore, I will not change my current ratings.

---

### Official Review · Reviewer_EXsm · 2023-07-06

**Soundness:** 3 good
**Presentation:** 2 fair
**Contribution:** 4 excellent
**Rating:** 7
**Confidence:** 2

**Summary:**

This work introduces the tempo of adaptation in a non-stationary RL problem. The authors provide Proactive Tempo control Model-based (PTM) framework, and two specific instances PTM-T and PTM-G. By adjusting the tempo of the algorithms, the proposed algorithm can match the tempo of the environment to address non-stationarity. This claimed property is demonstrated by both theoretical analysis and empirical results. The proposed PTM framework helps the RL algorithm to be implemented in real-world settings. The authors show that the PTM framework achieves a higher online return than the existing methods and provides empirical evidence of the existence of an optimal algorithm tempo with the comprehensive experimental evaluation of various Mujoco tasks.

**Strengths:**

(1) Interesting topic for real-world applications: This paper focuses on a practical setting for non-stationary RL by introducing a “time-elapsing variation budget”, which not only measures non-stationarity but also considers actual time taken. This setting is common and has great potential in real-world applications, and this work will encourage more interest in this direction.

(2) Solid theoretical analysis: This paper provides a dedicated mathematical formulation for the tasks and provides a detailed theoretical analysis and discussion of the proposed methods under the tabular settings. They also provide a discussion on more general cases in the supplementary material.

(3) Comprehensive experimental evaluation: This work provides comprehensive experimental results both with sufficient baseline methods and testing tasks to demonstrate the strength of their work.


**Weaknesses:**

(1) Complicated symbolic notation: The author uses too many fancy symbols for notations that are hard to identify, write, pronounce, and memorize. I would be more than happy to see a simplified version, and I believe by reducing the symbolic complexity, this paper will be much easier to follow.

**Questions:**



**Limitations:**

The authors adequately addressed the limitations

---

> ### Author Rebuttal · Authors · 2023-08-09
>
> $\textbf{1) W1: complicated symbolic notation}$
>
> Thanks for pointing out that the paper contains too many notations and those make readers follow up the paper hard. We will reduce notations and make theorems much more straightforward in the camera-ready version if we get accepted.
>
> $\textbf{2) Regarding with low-confidence score: clarification of main contribution}$
>
> Besides your constructive comments, we also appreciate your low-confidence score and we understand the score since the message of our paper is not fully clear to the reviewer. Within this sense, we would like to clarify the motivation, main contribution and how our theoretical analysis is related, and how our experiments support those.
>
> First, We would like to emphasize that our work’s $\textbf{main contribution}$ is interpreting the non-stationary MDP as “$\textit{wall-clock time}$” goes by, not as “$\textit{episode}$” goes by, and we claim shifting the perspective from episode to time is a more realistic setting that matches with non-stationary RL’s motivation: RL for real-world application. Raising this issue recalls our new problem setting stated in the introduction (line 2 $\sim$ 5). This new setting gives rise to solving an additional problem: how to choose the optimal $K-$length time sequence that the agent interacts with the environment?”. With this respect, our $\textbf{main theoretical result}$ is that we determine the optimal K-length interaction time sequence by finding an optimal training time of the agent. Within this sense, we propose the PTM framework to 1) find optimal training time by forecasting model-based method and 2) claim our method yields a better online return, which 1) is supported by Fig 3-(a) and 2) is supported by Table 1.
>
> Therefore, we claim  $\textbf{we unearth a hidden assumption}$ that the agent and the environment should have independent timelines, and our proposed method finds how to adapt two timelines theoretically and also proves empirically, which supports we have different problem-setting and different purpose of the algorithm aside from existing the non-stationary RL methods.
>
> We also brought up our motivation and main contribution on “global response”

---

> > ### Comment · Reviewer_EXsm · 2023-08-11
> >
> > Thank you for your detailed response and extra clarification on the contribution. Regarding W1, I agree that it would be beneficial to reduce the notation and make theorems more straightforward. After reading other reviews and the corresponding response, I would love to increase my score in favor of acceptance.

---

### Official Review · Reviewer_2jp7 · 2023-07-06

**Soundness:** 3 good
**Presentation:** 3 good
**Contribution:** 3 good
**Rating:** 7
**Confidence:** 3

**Summary:**

This paper presents a solution to nonstationary RL issues through a model-based framework called Tempo-Control (PTM). Specifically, the authors identify a new trade-off in nonstationary RL problems: the trade-off between learning an accurate model and learning an optimal policy. Based on this, a new framework is designed to allow the agent to find an appropriate tempo for adaptation. The authors provide theoretical analysis for the tabular version of the framework and a strategy for complex RL scenarios. Empirical evaluation results confirm the effectiveness of the scheme as well as its theoretical optimality properties. Overall, this paper offers insightful analysis on nonstationary RL problems and rigorous theoretical study on the method. I am inclined to give an "accept" rating.

**Strengths:**

[**About the framework's motivation**] The author's analysis of the trade-off between adaptation tempo and environment tempo is quite insightful. Additionally, the critique of the existing three research lines in nonstationary RL could provide inspiration to the field of nonstationary RL or RL in general.

[**About the theoretical analysis**] The paper offers rigorous theoretical analysis for the tabular version, providing solid theoretical guarantees for subsequent applications.

[**About method design**] The modeling method for adaptation tempo and environment tempo is clear and simple, offering strong reproducibility and portability.

[**About experiments**] The experiments use common nonstationary (deep) RL benchmarks, and the results are quite good, with a significant margin over the baselines. Furthermore, the ablation studies also confirm the previous theories on optimality.

**Weaknesses:**

Note: Most of the weaknesses listed below are more like questions, discussion points, or suggestions, rather than outright flaws. Any clarifications provided by the authors would be very welcome and appreciated. Given the limited time for rebuttal, it is not necessary to fully supplement experiments for comparison with these papers. Some clarifications and discussions on these methods would be greatly appreciated.

[**About the latent variable**] The authors assume that the non-stationary variable $\hat{o}_k$ is observable, but in real complex scenarios, it is challenging to observe the non-stationary variable. Additionally, certain specific assumptions are required about this latent variable (such as the function of the latent variable's change with time, dimensions, etc.). Could the authors provide more clarification on this point?

[**About the presentation**]The authors might consider placing PTM-G after the third section and moving Algorithm 3 from the appendix to the main text. This could help readers better understand the overall framework design and flow.

[**About the experiment**] In Fig. 3 (b-2), it seems that convergence has not been reached at the end; consider providing a longer range for the x-axis. Also, the x-axis is not labeled in Fig. 3.


**Questions:**

I have included the questions in the above section.

**Limitations:**

The authors have given a detailed analysis of the limitations in the paper.

---

> ### Author Rebuttal · Authors · 2023-08-09
>
> $\textbf{[About the latent variables : assumption validity]}$
>
> Thanks for pointing out the validity of the assumption.
>
> First, we would like to mention the biggest reason for the observable $\mathcal{O}$ assumption is not to increase the gap between theoretical analysis (PTM-T framework) and solid experiment results (PTM-G framework). Please note that most relevant existing works train a probabilistic network that encodes the $1 \sim k$ episodes’ trajectories  (which represents the $1 \sim k$ episodes’ environment)  into the latent space (which represents the non-stationary variables), and learn network parameters by bayesian method, then infer the future non-stationary variable from the latent space [9-12]. The problem with Bayesian inference is that its posterior distribution is intractable. Within this sense, We have not used the existing method since the intractability of the practical algorithm makes the gap between the theoretical analysis and the practical algorithm larger.
>
> Also, we would like to recall that assumption 1 is not necessary for the PTM-T framework, which means we $\textit{estimate}$ the non-stationary variables, then predict the future variables.  (please refer to line 186 $\sim$ 187). The practical algorithm, PTM-G framework, exploits the advantage of assumption 1 to forecast the future variables (future MDP model) to fully support the ablation study. Throughout the paper, we stated the theoretical analysis on “improving forecaster accuracy”(section 4.1.2) to emphasize that “MDP forecaster” serves a significant role in PTM-framework, and have done experiments to support its theoretical importance (Figure 3-(b-1), (b-2)). We have thought inferring the non-stationary variables makes the ablation study that checks the performance of the MDP forecaster to be unclear (Figure 3-(b-1), (b-2)) since it takes $w$ past observations.
>
> $\textbf{[About the presentation]}$
>
> Thanks for pointing out the location of the PTM-G algorithm. We will modify it later on if we get accepted.
>
> $\textbf{[About the experiment : Figure 3-(b-2) issue]}$
>
> Thanks for pointing out the issue of convergence. Does to the limited time of the rebuttal, we have not attempted to do additional experiments. However, we can certainly claim that we have rolled 200 episodes, and the model error except f=ARIMA increases as the episode goes by.

---

> > ### Comment · Reviewer_2jp7 · 2023-08-11
> >
> > Thanks for the detailed response. Most of my concerns have been addressed. I think it would be better if the authors could include the explanation of the validity of the assumption in the future version.
> >
> > Meanwhile, after reading the other reviews and the corresponding rebuttals, I would keep my original rating. As to the experimental part, I think the current baselines and benchmark choices have been already somewhat convincing to show the effectiveness of this approach.

---

### Official Review · Reviewer_4AXz · 2023-07-25

**Soundness:** 3 good
**Presentation:** 2 fair
**Contribution:** 3 good
**Rating:** 7
**Confidence:** 3

**Summary:**

This paper proposes to look at non-stationarity in reinforcement learning (RL) from a new point of view. While previous approaches did not generally consider the practical implications of the time elapsed while learning a policy, the idea of the current paper is to consider exactly these implications.

A trade-off is posed between spending time learning a stronger policy versus collecting more samples and learning a better model. After defining the problem, a practical solution is proposed. A forecaster fits a model of the process driving the non-stationarity, as well as the set a model of Markov decision processes (MDP) conditioned on it. An Rl algorithm, potentially any algorithm, is then plugged to the forecasted MDP and is optimised for several iterations.

A theoretical analysis is conducted to demonstrate that, as anticipated, the regret drastically depends on the number of policy iterations. Fortunately, under certain conditions on the MDP and the algorithm, it is shown that a sub-linear regret can be obtained in the proposed framework. It results that there exists an optimal number of iterations which depends on the MDP itself.

Finally, an empirical study demonstrates that a practical algorithm, based on the previous theory, is able to compete and outperform several baselines, in a range of non-stationary problems. Furthermore, an ablation study is proposed to illustrate the theoretical results.

**Strengths:**

Despite my rating in "contribution", this paper comes with an interesting idea.
In a practical problem, the time spent while learning a policy may be time not spent to sample new data.  It is an interesting theoretical perspective. As expected, the authors demonstrate that it is critical to take this time into account. I believe that this theory could have great implications and am not aware of previous works going in the exact same direction.

The practical algorithm follows a simple idea, which I like. It comes with both theoretical and empirical achievements.

The theoretical analysis is helpful and and well conducted. I have not checked demonstrations in detail, but many proof steps are classic in the literature.  The conclusion of the theoretical analysis is great: "tempo" in the MDP and in the algorithms should be in harmony.

The experimental section is concise but holds enough value to underline the validity of the framework.

**Weaknesses:**

My rating for the "contribution" results from my doubts about the definition of the framework. Despite originating from an interesting idea, I question the following points:
 - The non-stationarity is not defined clearly enough. An agent is placed in an MDP where it can either sample a trajectory or train its policy. Why wouldn't the MDP change during the trajectory? If the agent can choose when to sample data in a stationary MDP, and that the non-stationarity is bounded as in line 119, it seems that too much control over the non-stationarity is offered to the agent.
 - The terms B_p(G) and B_r(G) are not given more details in the main paper. I understand that they satisfies the equation of line 119, but what is the meaning of Delta_t=1? This is also important to understand the experiments, is a step of non-stationarity corresponding to Delta_t=1 and therefore to an iteration of NPG?
 - I am concerned about the result of Theorem 2. I understand that, for a fixed K, there should be an optimal tempo for the policy iteration. But I miss a result for an agent that would have a budget in terms of time. How does a well trained agent sampling less trajectories compare to a poorly trained agent sampling more trajectories in the same time? The optimal tempo could be impacted by these considerations.
 - A related comment about the regret. In the regret formula, the reference is the optimal value function in MDP k, that is, at sampling times synchronised with the algorithm. What if the algorithm is compared to an optimal policy at different tempo ?
 - These two previous comments can be gathered as follow: what if the agent is compared to any agent acting in the same amount of time?
 - I expect that the training takes more time as more data is collected. Delta_t could refer to the longest time of them all, but how can the agent be sure that Delta_t has elapsed always before sampling? This is related to my comment on the agent having an impact on the non-stationarity of the environment.
 - When the agent optimises its policy, as explained in 3.2, paragraph 2), a synthetic trajectory is rolled out. What if the rollout has a comparable time with respect to sampling a trajectory. Shouldn't the number of rollouts add to the time elapsed? In expectation it is irrelevant but it becomes necessary for a high-probability results. Similarly, why is the result of Proposition 3, case 1, suggesting G=\infty? Only a single trajectory is needed?

**Questions:**

- Why is the framework considering non-stationarity inter-episode and not intra-episode?
 - Proposition 1 offers a sublinear regret with a condition that binds H and epsilon, this doesn't seem great for applications where H is fixed beforehand. What do the authors think about it?
 - What does \Delta_t=1 correspond to in the experiments?
 - Is there a simple way to compute a regret that compares to the best algorithm given a fixed time elapsed instead of number of episodes (K) fixed?
 - What would be the optimal number of iterations in that case?
 - In the experiments, why is the performance compared over the last 10 episodes only? What about the regret throughout learning?

**Limitations:**

Extra:
 - the text in figures 1.c) and 1.d) i9s too small
 - type line 302: "the state-of-the-art model-based model-free algorithm"

---

> ### Author Rebuttal · Authors · 2023-08-08
>
> Thanks for your constructive comments and the followings are our answers to the weakness, questions that the reviewer raised.
>
> $\textbf{1-1) W1,Q1:  validity of inter-episode changing MDP}$
>
> We have defined the non-stationary as changing across the episode and keeping stationary during the episode (line 99). We have two reasons why we think episode-varying MDP is still reasonable for our paper. First, episode-varying MDP could be regarded as a discrete change for a certain interval for an infinite Horizon case, which supports our work still solves non-stationary environments. Second, most related theoretical works assumed step-varying MDP, but a lot of related empirical works on complex non-stationary environments assumed MDP changes across the episode. For more details, please refer to recent non-stationary RL papers that change inter-episode : [9,11,13,18].
>
> $\textbf{1-2) W1 : Issue of $B_p(c\Delta_t)=c^{\alpha_p}B_p(\Delta_t)$ provides too much control on the agent}$
>
> The environment property (line 119) is the assumption that is given as prior, not a property that the agent can control ($c, \alpha_p$ is given ,not choosen by agent). The only parameter that the agent choose is $\Delta_t$, which represents a real-world sceniro that the agent learns the environment model based on observations only from $K$ interaction times $[t_1,t_2,..,t_K]$ where $\Delta_t = t_{k}-t_{k-1}, \forall k \in [K]$. Different interaction times yields different observations, which leads to different learned model.
>
> $\textbf{2) W2,Q3 : lack of details on time elapsing variation budget, meaning of $\Delta_t=1$}$
>
> $\Delta_t=1$ means the agent executes an episode for every one second, which means (k)th episode and (k+1)th episode’s interval is a second.In Figure 3-(a),  $\Delta_t = [1,2,3,4,5]$ means how much the agent skips the environment’s episode (we have added additional information in line 325~326). Also $\Delta_t$ means training time which is proportional to gradient steps of NPG, we set $\Delta_t=1$ corresponds to $38$ iterations.
>
> $\textbf{3) W 3,4,5 : the clarity of theroem2}$
>
> Before, we emphasize our work’s main contribution is interpreting the non-stationary MDP as “wall-clock time” goes by, not as “episode” goes by, and shifting the perspective from episode to time is a more realistic setting that matches with non-stationary RL’s motivation: RL for real-world application. Raising this issue recalls our new problem setting stated in the introduction (line 2 $\sim$ 5). This new setting requires to solve an additional problem: how to choose the optimal K-length time sequence that the agent interacts with the environment?”. With this respect, our main theoretical result is determining the optimal $K-$ time sequence by optimal training time, and we propose a PTM framework to find optimal training time.
>
> Now, we answer your understanding of “fixed K, the optimal tempo exists”. Yes, the optimal tempo exists, but it's not for the fixed episode(K), but for fixed “time duration”. For better understanding, recall the example of the introduction. Let’s say for fixed time duration 0[s]$\sim$15[s], robot A executes episodes for every 1 second, [0,1,..,15], and between time $t,t+1$, the agent trains the policy. Robot B executes episodes every 2 seconds [0,2,..,14] and between time $t,t+2$, the agent trains the policy. Let’s assume one policy update takes 1 second. Then, robot A can update one time, and robot B can update two times. Robot A interacts with the environment for 15 episodes and updates policy one time, which has more information about the environment but has an uncertain approximate optimal policy. Robot B interacts with the environment for 7 episodes and updates the policy two times, which has less information about the environment but has a better approximate optimal policy than robot A.
>
> $\textbf{4) W 6: relationship between training time and the amount of data}$
>
> We want to answer data amount in Buffer does affect training time. We have defined training time as the time that takes one iteration of training, and in theoretical analysis, one iteration of policy update is same as one gradient step of the NPG algorithm. We partially admit that the batch size of the trajectory data could affect the “policy update for one gradient step”, which is same as B_p(1) and B_r(1), but “G” policy iteration does not change if we fix the batch size of the trajectories for policy update.
>
> $\textbf{5) W7: including rollout time to traning time, the meaning of $G=\infty$}$
>
> We have included the sample complexity (in Appendix: Lemma3) and the approximation gap $\delta$ due to a finite data sample. Line 216 $\sim$ 217 shows how Lemma3 interleaves with our Propositoin1. Small $\epsilon$ enables tighter upperbound for $R_{Alg_\tau}$, but also requires the approximation gap $\delta$ to be small. Small approximation gap $\delta$ requires large sample complexity by Lemma3. This supports the point that you raised: “In $\sim$ results.”. However, we have not included “data collection time” in the training time, but want to claim that it does not harm our works since we have defined the training time as the time that consumes one policy iteration.
>
> Also, note that $G=\infty$ for the case of max(alpha_r, alpha_p) =0 matches with our intuition, since max(alpha_r, alpha_p) =0  corresponds to “stationary environment” (line 120 $\sim$ 121: typo on the paper, \alpha_p=\alpha_r=0 is stationary environment) and within the stationary environment, large policy iteration guarantees closer optimal policy.
>
> $\textbf{6) Q2}$
>
> In proposition 1, we think word “given” in line 213 makes misunderstanding, What we meant by was “we choose \epsilon >0 that satisfies H > (terms)”. Therefore, H is fixed ever since the MDP is given, and for given H, we choose \epsilon.
>
> $\textbf{7) Q4,5}$
>
> Definition of dynamic regret requires obtaining the optimal policy at each episode. In complex environment, it is hard to define what is “optimal policy”.

---

> ### Author Response · Authors · 2023-08-16
> **Looking forward for the feedback!**
>
> Dear Reviewer 4AXz,
>
> We first thank you for the positive feedback on our work's idea provided in the initial official review.  Since the discussion deadline is approaching, we wonder whether our above response to your initial review were fully addressed to your concerns. We appreciate your comments that a low rating comes from some doubts about the definition of the framework, and we believe we have addressed your doubts and concerns clearly in our rebuttal. Besides, we also highlight our work's main contribution and motivation for you to accelerate resolving your doubts about our framework.  We also believe the reviewer raised a constructive issue (trajectory sampling time) which is an interesting point, and we hope to have further discussions on this.
>
> __We wonder whether our comments are fully addressed to you, or if not, we would love to hear back from you!__
>
> Best regards,
>
> Authors

---

> > ### Comment · Reviewer_4AXz · 2023-08-16
> >
> > Thank you for the detailed answer.
> > It resolves some of my doubts but there remain some.  I comment all answers.
> >
> > 1-1) I have taken note of the authors answer to my comment on the exact definition of the non-stsionarity. It seems to me that something is missing. I read the authors answer to another reviewer, recalled here: "First, We would like to emphasize that our work’s  is interpreting the non-stationary MDP as “wall-clock-time” goes by, not as “episode” goes by, and we claim shifting the perspective from episode to time is a more realistic setting that matches with non-stationary RL’s motivation: RL for real-world application".
> > I believe that this is a refreshing idea with interesting implications, but it seems that the idea wasn't applied fully.
> > Why would an infinite horizon process suffer from non-stationary shocks exactly when an episode ends while it remains deterministic during an episode? This doesn't seem realistic to me.
> >
> >
> > 1-2)
> > It was clear to me that the non-stationary dynamics (directed by c and alpha_p) are not controlled by the agent.
> > My doubt was related to how the agent, in that setting, could "choose" which non-stationary MDP to sample from. Imagine a scenario where the agent select to spend 1 amount of time to learn a policy for 1 iteration. Then Delta_t should be one. At that point, the agent interacts with stationary MDP M_1 for the duration an episode.
> > If instead the agent choses to wait an extra 0.1 amount of time before sampling; the stationary environment that the agent is now sampling from is M_{1.1}. This has an impact the resulting return and learned model as well.
> >
> > 2)
> > It seems that you set Delta_t =1 to be one second, as well as the time it takes to sample an episode as well as 38 iterations of NPG. I think it should be set to one of these and the rest should be deducted.
> >
> > 3) There may be a misunderstanding of the setting on my side. But changing k or Delta_t should change the total time elapsed.
> >
> > The authors have addressed my concerns in answers 4), 5) and 6).
> >
> > 7)  I agree that this is a more complex quantity but do the authors but I believe that this is a quantity that we would like to compare to.

---

> ### Author Response · Authors · 2023-08-18
> **Thanks for the fast response and further classification.**
>
> Dear Reviewer 4AXz,
>
> Thanks for the response and helpful feedback. We made further clarification on your comments.
>
> $\textbf{0) wall-clock time consumption when the agent takes steps in an episode}$
>
> __To the best of our knowledge, we guess the critical misconception comes from "consideration on wall-clock time consumption when the agent takes steps (when rollouts a trajectory in an episode)"__. Throughout the paper, we __have not considered__ the time consumption when the agent takes steps in an episode. Our works are based on the assumption that one step takes an infinitesimal second. Namely, $t=0$, the agent starts 1st episode in $\mathcal{M}[t=0]$ and take $H$ steps. After taking $H$ steps, the agent still locates in $ \mathcal{M}[t=0] $ since we assume rollout time is infinitesimal. Then the agent starts to train the policy. If $\Delta_t=4$, then at $t=4$, the agent starts a 2nd episode at $\mathcal{M}[t=4]$.  We believe the reviewer has assumed that the time consumption to collect samples in an episode is proportional to step $H$.
>
> Nevertheless, we claim  __constant time-consumption when taking steps does not harm our work__.
> This is because the horizon $H$ is given from the environment, then the time to collect sample in an episode is given as constant as prior (let’s assume 2 [sec]  to finish a trajectory ($H$ steps =2 secs)). The 2 seconds delay that accumulates for every episode does not harm our theoretical analysis and practical experiments since all we need is resizing policy training time $\Delta_t \leftarrow \Delta_t +2 $.
>
> $\textbf{1-1) How does inter-episode changing MDP matches up with the paper's infinite horizon problem setting?}$
>
> Our problem is based on __finite__ (not infinite) horizon __inter-episode changing__ MDP where horizon is __fixed__  for all MDPs. (We have stated in the section 2 - line 95 $\sim$ 100) .
>
> For further clarification, we emphasize that our analysis is built upon finite horizon $H$.
> Specifically, in experiments, $\texttt{PTM-T}$'s tabular environment is based on $H=13$ (line 477 of Appendix A.1-1., and $\texttt{PTM-G}$'s Mujoco environment is based on $H=100$ (lined 1026 of appendix E.2 )
>
> $\textbf{1-2) How agent can ``choose'' which non-stationary MDP to start an episode?}$
>
>  __The agent can "choose’’ (or more correctly "pre-determine") the policy training time $\Delta_t$ based on Proposition 3__. The proposition 3 enables agent to pre-determine optimal training time $\Delta^*_t (=G^*)$, then pre-determine optimal interaction time sequence $t^*_k=t^*_1+(k-1) \Delta^*_t$ for all $k$ before starting a 1st episode at $t=t^*_1$. Then, agent starts an episode 1,2,..,K at time $t^*_1,t^*_2,..,t^*_K$.
>
> Intuitively, choosing the interaction time sequence __as prior__ is possible since the agent knew the ``time-elapsing variation budget’’ as prior information. Please note that existing works also assume the agent knows a variation budget as prior [13-17,22,37,38] to compute the dynamic regret. Also, Proposition 3 needs $B_p(1),B_r(1)$ to compute  $\Delta^*_t (=G^*)$.
>
> $\textbf{2) Issue of $\Delta_t$ = 1}$
>
> In theoretical analysis, we set policy training time $\Delta_t$[sec] equal to policy iteration number $G$ [EA].
> In the experiment, we set 38 policy iterations as equal to one second, which means $\Delta_t = \lfloor G / 38  \rfloor$.
>
> $\textbf{3) Issue of changing $K$ or $\Delta_t$ changes total time elapsed} $
>
> Changing $K$ or $\Delta_t$ __does not change__ the total time elapsed. Please recall the robot example in the above answer to W3. __For fixed time duration [0,T]__, the agent can choose different training time(= differnt $K$). Also, regardless of how the agent chooses $K$, the sublinear regret is guaranteed by Proposition 1,2,3.
>
> $\textbf{4) Showing the whole training procedure as a dynamic regret}$
>
> We first note that most existing empirical non-stationary RL works on __high-dimensional MDP__  $[9,10,11,14]$ shows the algorithm performance by the __average reward return__. Namely, they contain $\textcolor{red}{(1)}$ Figures of whole training time: average reward (y-axis) / total episode (x-axis) and $\textcolor{blue}{(2)}$ A Summary Table: Run multiple experiments with different seeds, then take an average of a return (from last episode) over multiple seeds. Also, some existing empirical works on non-stationary __Bandit setting__ $[13]$ show the algorithm performance by a __regret__.
>
> __Our work's environment is high-dimensional MDP, and we have a result $\textcolor{red}{(1)}$ as Figure 5 $\sim$ 13 in Appendix, and a result $\textcolor{blue}{(2)}$ as Table 1 in the main paper__.
>
> Also, to answer W7 - why take last 10 episodes, Our Table 1 shows result that first takes the average over the final returns of the last 10 episodes, then takes the average over multiple runs. Rather than showing only the last episode's result on a table, we used our metric to show a more reliable result, since its learning continuously in a non-statioary environment.

---

> > ### Comment · Reviewer_4AXz · 2023-08-19
> >
> > It seems that we are mis understanding each others. I have not assumed that the time consumption to collect samples in an episode is proportional to step $H$. This is clear to me. However, my doubt is that, in a real non-stationary scenario, the environment could also change during an episode, and that asks some questions about the current framework.
> > Let me do an example, assume, that in practice, the sampling time for one episode is 1 second. Let $t_1$ be the real time elapsed and $t_2$ be the time elapsed as in your framework. When $t_1=1$, the sampling is done. The training of the policy starts and $t_2$ is still 0. I imagine that equation (2.1) applies only to $t_2$ in your framework but, for a real dynamical system, it would apply to $t_1$. Then, since 1 second as elapsed in the real time, the environment could possibly have changed during sampling. My doubt is:  why can we consider the environment stationary during an episode and only non-stationary in between episodes?
> >
> > we are also misunderstanding on the total time elapsed. In your robot example, I understand that robot B is sampling only half as much. But both robot interact $K$ times. Therefore robot A finishes its sampling and training at time $K$ while robot B finishes at time $2K$. Isn't it?

---

> ### Author Response · Authors · 2023-08-19
> **Further clarification on misundestandings - comment 1**
>
> Dear Reviewer 4AXz,
>
> Thank you for your prompt response!
>
> We genuinely appreciate your effort in clarifying the misunderstandings. Your detailed feedback has enabled us to address the concerns more accurately. We would be grateful if you could confirm whether the issues you raised in points 2), 3), and 4) of your previous comment have been addressed satisfactorily. If not, we are more than willing to provide further clarification.
>
> $\textcolor{red}{\text{For clarity, we broke down your comments into comment 1 (first paragraph) and comment 2 (second paragraph)}}$
>
> $\textbf{[Comment 1-1] Different timelines between real-time and framework. How do you apply to equation 2.1?}$
>
> We recognize the discrepancy between the time elapsed in real-time, denoted as $t_{real}$, and the time within our framework, denoted as $t_{frame}$. Let's define the sampling time for a single episode as $\Delta_{sp}$ and the training time as $\Delta_{t}$. Initiating the system with $t_{real} = t_{frame} = 0$, the agent starts the first episode immediately. When the agent embarks on the second episode, $t_{real}$ is $\Delta_{sp} + \Delta_{t}$ while $t_{frame}$ is $\Delta_{t}$. Extending this logic, for the $k^{th}$ episode, $t_{real}$ is $(k-1)\Delta_{sp} + (k-1)\Delta_{t}$ and $t_{frame}$ is $(k-1)\Delta_{t}$.
>
>  This observation is indeed pertinent. To rectify it within our framework, we can adjust the time discrepancy. Specifically, we can modify $\Delta_{t} \leftarrow \Delta_{sp} + \Delta_{t}$. When agent determines $\Delta_{t}$, it should incorporate the constant term $\Delta_{sp}$, which is given as environment information. This assumption is realistic since $\Delta_{sp}$ can be estimated based on horizen $H$ and the duration of each step, including when the agent acts, the environment reacts, and feedback is offered.
>
> An implication of this is in the optimization of the update tempo (e.g., $G(=\Delta_t)$) in Proposition 3. By the definition provided in our paper, which solely considers the training time $\Delta_t$, we should replace $\Delta_{t}$ with $\Delta_{sp} + \Delta_{t}$. This results in a simple modification to the optimal $G$ in Proposition 3, producing a lower bound on $G^*$, that is, $G^*_{Alg} = \min ( \sqrt{k_{Alg} / k_B} , \Delta_{sp})$.
>
> $\textbf{[Comment 1-2]My doubt is: why can we consider the environment stationary during an episode and only non-stationary in between episodes?}$
>
> We first note that our MDP setting (piecewise nonstationary MDP) is also utilized in existing non-stationary RL work $\textbf{[9]}$(Chen et al. 2022). Also, this is an important question that touches on fundamental assumptions that underpin our work. We elaborate as follows:
>
> - Just as continuous signals in signal processing are transformed into discrete steps through quantization, we adopt a similar strategy. There is no difference in terms of accumulated change within a given period, but there may be difference within each sampling interval, where the analog signal changes continuously but the discretized signal is held constant. Our model mirrors this by transferring the accumulated change from one episode to the next, thereby treating individual episodes as quasi-stationary.
>
> - In signal processing, discretization's validity hinges on the sampling frequency outstripping the signal change rate. In our context, this translates to the environment being relatively stationary throughout an episode's real-time span, though changes accrue and carry over to subsequent episodes.
>
> - Our assumption stands because controllability is often feasible. Consider high-frequency trading: with a trading capability of once every second and a MDP horizon of 10, the sampling time spans 10 seconds—a period where markets might witness substantial shifts. Contrastingly, if trades occur every millisecond or even faster, a 10-step MDP horizon implies a mere 10 milliseconds of sampling time. Here, expecting the market to remain stationary is more reasonable.
>
> - The MDP formulation, influenced by numerous external variables like actuation frequency, precedes algorithmic design. Provided the environment under study remains relatively stationary during our sampling time per the MDP formulation, our algorithm remains applicable. The foundational assumption is that, within each real-time MDP episode, the environment retains its stationary nature.
>
> - While we treat each episode as stationary, this doesn't negate our engagement with nonstationary environments. Cumulative changes, in effect, shift to the next episode, much like how signal discretization functions. As long as the "baseline resolution" (i.e., the real-time duration of an episode) is relatively brief compared to the intrinsic tempo of the environment, this presumption stands firm. Empirically, our experiments validate this assumption does not hinder performance across the tested environments.
>
> - Though intriguing, the co-design of environment with algorithm remains a complex domain for future exploration.

---

> ### Author Response · Authors · 2023-08-19
> **Further clarification on misundestandings - comment 2**
>
> $\textbf{[Comment 2] Misunderstanding on the total time elapsed}$
>
> Thank you for highlighting this matter again. We've expanded upon the robot example to clarify any misunderstandings. Given a fixed real-time duration $t_{real} \in [0,T]$, let's define the sampling time as $\Delta_{sp}$. Suppose robot A has a training policy duration $\Delta_t$ (denoted as $\Delta_A$), and robot B's is twice that, i.e., $2\Delta_t$ (denoted as $\Delta_B$).
>
> The number of episodes for robot A is then calculated as $K_A = \lfloor T / (\Delta_t + \Delta_{sp} ) \rfloor$ and for robot B as $K_B = \lfloor T / (2\Delta_t + \Delta_{sp} ) \rfloor$. Notably, the interaction count (number of episodes) varies: $K_A > K_B$, but the policy training time is inversely related, with $\Delta_A < \Delta_B$.
>
> We wish to underline that this __inverse relationship__ is pivotal. It brings forth the trade-off dilemma and illuminates the quest for an optimal training duration, culminating in Proposition 3.

---

> > ### Comment · Reviewer_4AXz · 2023-08-20
> >
> > Dear authors,
> >
> > I confirm that your answer in point 4 addresses my concerns. The one in point 3 is related to my comment (Comment 2 as you have called it). Point 2 partially addresses my concern but I it is not critical in my evaluation of the paper.
> >
> > Regarding your answers to my latest comments,
> >
> > Comment 1-1: I appreciate the clarification of the framework and believe that it could help readers that, like me, find that the framework lacks some realism.
> >
> > Comment 1-2: I am aware that certain related works consider piecewise non-stationary MDP but the novelty of your framework is to consider wall-clock-time. Therefore, I consider that, if non-stationary is not modelled at the inter-episode level, a discussion should be provided for why this is the case. I like the examples that you have given with the discretisation and think that it would find its place within the paper. I also note the example on high-frequency trading and think that this could be the occasion to warn the reader that the framework is suited for environment where $\Delta_sp$ is negligeable compared to $\Delta_t$.
> >
> >
> > Comment 2: I understand that the bound on the number of the difference results depend on $K$ but I assumed this quantity fixed. It seems to be nowhere indicated that K is a variable. Sentences such as  during "the total K episodes" (l. 99) or " Then, at the end of episode $k \in [K]$" (l. 133)  mislead me. Moreover, as I understand proposition 3, it gives, for a fixed $K$, the optimal number of policy iteration. It doesn't seem to answer the question: what is the optimal number of policy iteration given that I have a time budget of $T$?  Furthermore, for a fixed K, changing $G$ changes the real time elapsed. It seems that this bounds, by considering different values of G for a fixed $K$, compares the regret of different process that are at a different point in time. In addition, the definition of $B_p(\Delta_t)$ and $B_r(\Delta_t)$ depend on $K$ but it is not explicit from the notation. This adds to the belief that $K$ is a constant.
> >
> > The authors have satisfactorily addressed one of my two main concerns. I note that the other reviewers are leaning toward the positive side. I am happy to follow their decision and raise my score. Note that I still consider my comment 2 to no have been solved yet.

---

> ### Author Response · Authors · 2023-08-21
> **Further clarification on comment 2**
>
> Dear reviewer 4AXz,
>
> Thanks for the constructive and insightful feedback, which helped us improve the quality of the paper.
> We have provided further clarification on comment 2.
> Although the discussion period is closing soon, we welcome the reviewer's further concerns anytime and we will try to answer as fast as possible.
>
> $\textbf{[Comment2] Misunderstanding on the total time elapsed}$
>
> Thanks for re-raising this issue. We agree that the reader can naturally assume $K$ to be fixed while going through the paper and stating "variable" $K$ would not mislead the reader to misunderstand the paper. We will clearly add that $K$ is the variable that the agent should choose beforehand in the camera-ready version if we get accepted.
>
> However, It is straightforward to change the regret bound depending on $T$, since $K$ and $T$ are in the relationship with a closed form, i.e. $K := \lfloor T / (\Delta_{sampling} + \Delta_{training})\rfloor$[Eq (1)]. Please note that inserting Eq (1) into all theorems, propositions of the paper also guarantees sublinear in $T$ of their regret bounds. This supports that substituting all theorems written on $K$ to $T$ does not harm our works. Also, this leads to straightforward answers to the reviewer's additional concerns as follows.
>
> $\textbf{[Comment2-1] It doesn't seem to answer the question: what is the optimal number of policy iterations given that I have a time budget of $T$}$
>
> __Optimal number of policy iterations does not change__ . Intuitively, this is because it depends on how fast the environment changes ($B_r(G),B_p(G)$) and environment (or MDP) constants ($\alpha_r,\alpha_p$) as stated in Proposition 3. Namely, for a given $K$, let its  optimal number of policy iterations as $G^*$. Then for any $T \in [K(\Delta_{sampling} + \Delta_{training}),(K+1)(\Delta_{sampling} + \Delta_{training}) )$ also have same $G^*$.
>
> $\textbf{[Comment2-2] Why theoretical works on based on $K$, not $T$}$
>
> The main reason why we stated all works on $K$ is to emphasize __regardless of how the agent choose $K$, we can compute the optimal $G^*$ while also guarantee sublinear regret in $K$.__
>
> We also made some further clarification on some sub conserns in comment2 as follows.
>
> - __Moreover, as I understand proposition 3, it gives, for a fixed $K$__ $\rightarrow$ What we meant by fixed $K$ is that $K$ is pre-determined by the agent, so it is fixed with respect to the agent during the whole time $t \in [0,T]$, not $K$ is given as constant.
> - __Sentences such as during "the total K episodes" (l. 99) or " Then, at the end of the episode " (l. 133) mislead me__ $\rightarrow$ Same reason with above. "total K episodes, end of the episode" are all respect to the "agent" since  the agent executes the 1st episode after determining what is optimal $K$ (= optimal $G$)
> - __The definition of time elapsing variation budget depends on K__ $\rightarrow$ Yes, this is true. For fixed time duration $[0,T]$, the agent can choose $K$, and interaction time instances {$ t_1,t_2,..,t_K $}. Depending on how many times the agent interacts with the environment (=$K$) gives a different evaluation of environment non-stationarity(=timp elapsing variation budget). This is well-elaborated by our robot example.

---

> > ### Comment · Reviewer_4AXz · 2023-08-21
> >
> > Dear authors,
> >
> > You have addressed my last concern with this answer. I will align on the acceptance side. I stress, however, that the authors must work on the clarity/presentation of the paper. This has caused me some misunderstandings and the issue has been raised by by the other reviewers as well.

---

### Author Rebuttal · Authors · 2023-08-09

We appreciate constructive comments from all reviewers.

We would like to organize our $\textbf{motivation and main contribution}$ on this global response to help reviewers to better understand our paper. We also have elaborated more details on each reviewer's responses.


$\textbf{1. Motivation}$

$\textbf{We unearth a hidden assumption of non-stationary reinforcement learning (RL)}$ that critically holds back its fundamental motivation: real-world applications of RL. We call it a $\textit{time synchronization}$ issue between the agent and the environment. In the real-world, the environment changes as a $\textit{wall-clock time}$ goes by, not as an $\textit{episode}$ goes by, where the $\textit{wall-clock time}$ should independently flow from the episode which is controllable by the agent. The $\textit{time}$ (wall-clock time) of the environment should be uncontrollable by any instances including the agent, training algorithm.

To solve this issue, we consider a $\textbf{practical setting}$ for non-stationary RL where the agent chooses a time sequence when to interact with the environment and fully spends the time interval(the time between $k^{th}$ and $k+1^{th}$ interaction time instance) to train its policy.


$\textbf{2. Main contribution}$

Our main contribution is that finding the optimal time sequence entails an additional factor, $\textbf{tempo}$, and an additional trade-off that should be addressed in addition to the traditional exploration-exploitation trade-off, that is $\textbf{the trade-off between the environment tempo and the agent tempo}$.  We propose a $\texttt{PTM}$ framework that finds optimal training time by leveraging how long the agent should train its policy (agent tempo) and how fast the environment changes (environment tempo). Theoretically, this work establishes an optimal training time as a function of the degree of the environment's non-stationarity and achieves a sublinear dynamic regret at the same time. Our experimental evaluation on Mujoco non-stationary environments shows that the \texttt{PTM} framework achieves a higher online return than the existing methods, and it provides empirical evidence of the existence of an optimal algorithm tempo.

---

### Decision · Program_Chairs · 2023-09-21

**Decision:**

Accept (poster)

**Comment:**

This paper introduces a new reinforcement learning framework to deal with non-stationary environments.
Although the reviewers raised many concerns about the presentation, the gap between theory and practice, and the experimental evaluation, the authors' rebuttals effectively solved most of these concerns.
Even if the reviewers did not reach a consensus, there is quite a strong support for acceptance.
The authors need to be aware that there were many problems with the presentation that induced a lot of misunderstandings and prevented some reviewers from properly evaluating their work.
To avoid the same problems with future readers of this paper, the authors have to deeply revise the presentation of their paper and fix the issues raised in the reviews.